# Modular, cascade-like transcriptional program of regeneration in *Stentor*

**Pranidhi Sood[1], Athena Lin[1], Connie Yan[1], Rebecca McGillivary[1], Ulises Diaz[1], Tatyana Makushok[1], Ambika V Nadkarni[2], Sindy KY Tang[2], Wallace F Marshall[1]\***

[1]Department of Biochemistry & Biophysics, University of California, San Francisco, San Francisco, United States; [2]Department of Mechanical Engineering, Stanford University, Palo Alto, United States

**Abstract** The giant ciliate Stentor coeruleus is a classical model system for studying regeneration and morphogenesis in a single cell. The anterior of the cell is marked by an array of cilia, known as the oral apparatus, which can be induced to shed and regenerate in a series of reproducible morphological steps, previously shown to require transcription. If a cell is cut in half, each half regenerates an intact cell. We used RNA sequencing (RNAseq) to assay the dynamic changes in Stentor's transcriptome during regeneration, after both oral apparatus shedding and bisection, allowing us to identify distinct temporal waves of gene expression including kinases, RNA -binding proteins, centriole biogenesis factors, and orthologs of human ciliopathy genes. By comparing transcriptional profiles of different regeneration events, we identified distinct modules of gene expression corresponding to oral apparatus regeneration, posterior holdfast regeneration, and recovery after wounding. By measuring gene expression after blocking translation, we show that the sequential waves of gene expression involve a cascade mechanism in which later waves of expression are triggered by translation products of early-expressed genes. Among the early-expressed genes, we identified an E2F transcription factor and the RNA-binding protein Pumilio as potential regulators of regeneration based on the expression pattern of their predicted target genes. RNAi-mediated knockdown experiments indicate that Pumilio is required for regenerating oral structures of the correct size. E2F is involved in the completion of regeneration but is dispensable for earlier steps. This work allows us to classify regeneration genes into groups based on their potential role for regeneration in distinct cell regeneration paradigms, and provides insight into how a single cell can coordinate complex morphogenetic pathways to regenerate missing structures.

**\*For correspondence:**
wallace.marshall@ucsf.edu

**Competing interest:** The authors declare that no competing interests exist.

## Editor's evaluation

This ground-breaking study builds on recent genome annotation to report the gene expression pattern that drives morphogenesis of Stentor, a large and beautifully organized single-celled organism with a remarkable ability to regenerate after damage. The study has been greatly strengthened by the addition of two molecular perturbation experiments that provide important insights into the regeneration process.

## Introduction

While much is known about the molecular composition of cells, the mechanism by which those components are arranged into complex patterns and structures is far less understood (*Kirschner et al., 2000*; *Shulman and St Johnston, 1999*; *Harold, 2005*; *Marshall, 2020*). How does a cell create and maintain pattern? Historically, study of regeneration has played a key role in revealing mechanisms of animal development, because regeneration allows specific developmental processes to be induced

experimentally (*Morgan, 1901a*). Understanding how cells are able to rebuild cellular components and re-establish global patterning holds the promise of shedding new light on the largely unanswered fundamental question of how cells perform morphogenesis and pattern formation.

A second reason to study regeneration in single cells is to identify the mechanisms by which cells repair and recover from wounds. Regeneration and wound healing are processes that are typically studied at the tissue level in multi-cellular organisms, but individual cells also must be able to repair wounds following mechanical disruption, and a cellular scale response to injury is a crucial feature of repair even in multi-cellular organisms. Injured cells must be able to not only patch over the site of injury to prevent leakage of cytoplasm, they also need to re-establish polarity, rebuild organelles, and reorganize the cytoskeleton (*Tang and Marshall, 2017*). Deficiencies in cell repair have become implicated in disease, for example in diseases of the heart, lung, and nervous system (*Oeckler and Hubmayr, 2008*; *Wang et al., 2010*; *Angelo and Mao, 2015*), including ARDS (acute respiratory distress syndrome) (*Cong et al., 2017*). Yet, there is still much to be learned about how an individual cell responds to a wound.

Ciliates provide an excellent opportunity for investigating the mechanisms of cellular patterning and regeneration due to the fact that they have highly stereotyped cell shapes, with easily visible surface structures that serve as landmarks to assess the progress of regeneration (*Aufderheide et al., 1980*; *Frankel, 1989*). The giant ciliate *Stentor coeruleus* (*Tartar, 1961*; *Marshall, 2021*) is a single cell that can fully regenerate its complex subcellular structure after injury. In this classical system, virtually any portion of the cell, when excised, will give rise to a normally proportioned cell with intact subcellular organization (*Morgan, 1901b*; *Tartar, 1961*). *Stentor* provides a unique opportunity to study regeneration and patterning at the cell scale. Its large size (up to 1–2 mm in length), clear anterior/posterior axis, detailed cortical patterning, and remarkable ability to heal even large wounds in the cell membrane make it especially amenable to surgical manipulation and imaging approaches. But despite roughly a century of analysis by microscopy and microsurgery, the mechanism by which Stentor forms and regenerates its complex body pattern remains a mystery, because a lack of genetic and genomic tools in the organism prevented detailed molecular analysis of its processes. Now that the *S. coeruleus* genome has been determined (*Slabodnick et al., 2017*) and RNA interference methods developed (*Slabodnick et al., 2014*), molecular and genomic approaches enable new ways to explore this classic model system. In *Stentor*, principles of single-cell wound healing and regeneration can be studied without confounding effects of surrounding cells that may non-autonomously influence an intracellular injury response in the context of tissues.

Transcriptome studies of various multi-cellular animal species, including zebrafish and planaria, all of which are capable of regeneration throughout life, have begun to reveal key regulators of multi-cellular regeneration. Many of these studies have delineated the molecular players in regeneration by identifying genes that are expressed when stem cells differentiate into various cell types required to rebuild lost tissue or organs. This transcriptomic approach has thus proven its utility in revealing cell-specific requirements for regeneration in the context of tissues. We have sought to take a similar approach to the problem of single-cell regeneration in *Stentor*, focusing on two regenerative processes: regeneration of the oral apparatus (OA) after its removal by sucrose shock, and regeneration of cells that have been bisected into two half-cells.

One of the most dramatic and tractable regeneration paradigms in *Stentor* is the regeneration of the OA. The OA is a prominent structure on the anterior side of the cell that contains thousands of basal bodies and cilia (*Paulin and Bussey, 1971*) organized into a ciliated ring known as a membranellar band (MB). At one end of the ring is an invagination of the plasma membrane, where food particles are ingested. This invagination together with its associated cytoskeletal structures is known as the mouth. The OA can be induced to shed using sucrose shock (*Tartar, 1957*) after which a new OA regenerates over the course of 8 hr, progressing through a series of well-characterized morphological stages (*Figure 1A*; *Tartar, 1961*). Removal of the macronucleus, at any stage, causes OA regeneration to halt at the next stage, suggesting that several waves of gene expression may be required to drive different processes at different stages (*Tartar, 1961*). In addition to the macronucleus, each *Stentor* cell contains multiple micronuclei, which are generally not visible because they are closely adjacent to the macronucleus. The micronuclei are required for sexual reproduction but are dispensable for regeneration and for mitotic cell division (*Schwartz, 1935*; *Tartar, 1961*). Chemical inhibitor studies showed that regeneration of the OA requires transcription (*Whitson, 1965*; *James, 1967*; *Burchill,*

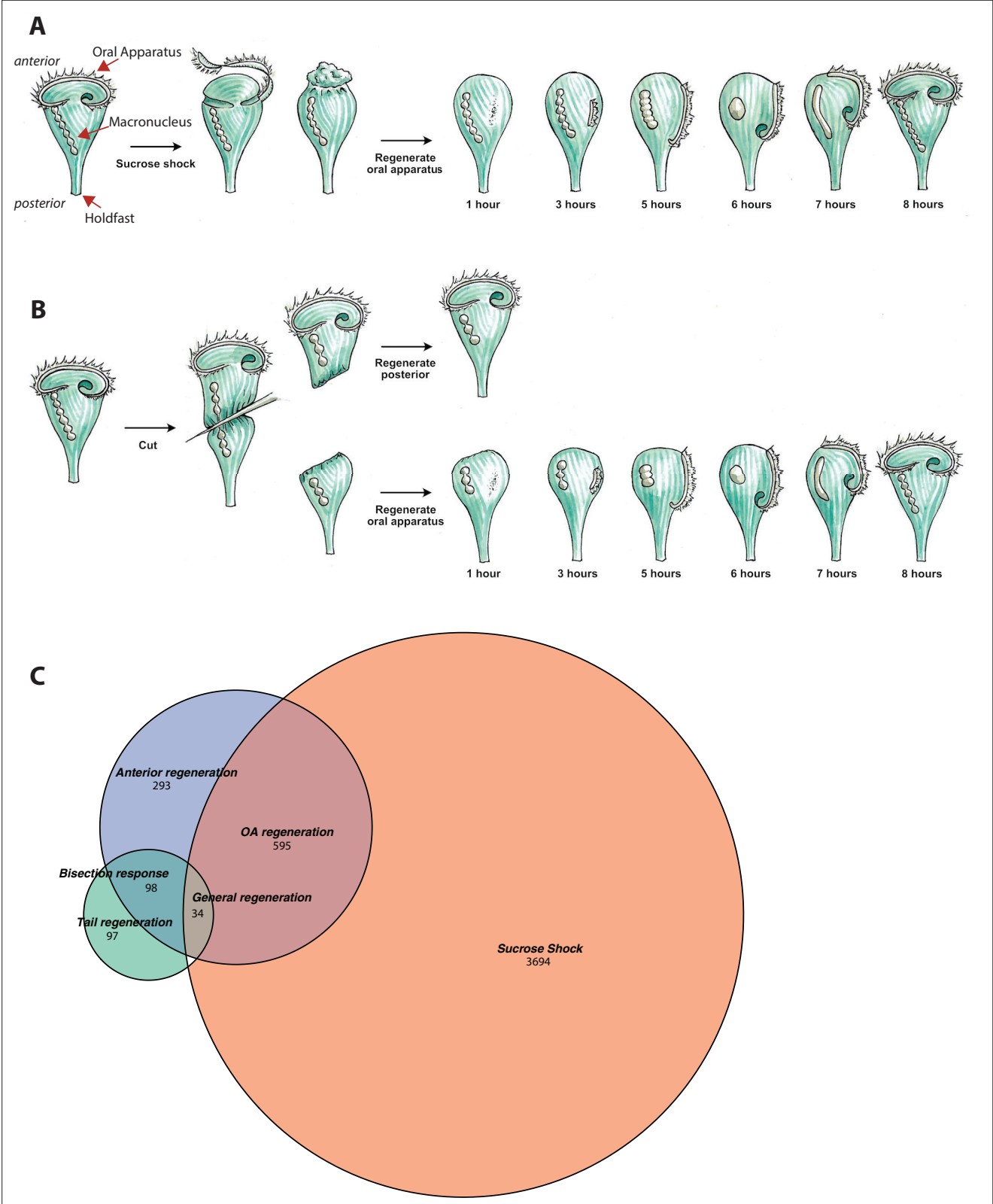

**Figure 1.** Transcriptional profiling of *Stentor* regeneration. (**A**) Morphological events in *Stentor* regeneration. At t=0,the membranellar band is shed during sucrose shock. The body cilia remain on the cell. After sucrose shock the frontal field protrudes, resulting in the anterior end of *Stentor* becoming rounded rather than cone-shaped. One hour after the start of regeneration, basal bodies begin to form at the locus of stripe contrast. After 3 hr, the first cilia of the new membranellar band are visible. These cilia show uncoordinated beating. After 5 hr, the new membranellar band elongates and

*Figure 1 continued on next page*

*Figure 1 continued*

extends along the anterior-posterior axis. A site for the new mouthparts is cleared at the posterior end of the membranellar band. During this stage, the cilia become oriented with respect to each other and their beating begins to become coordinated, forming multiple short metachronal waves. The nodes of the macronucleus begin to condense. By 6 hr, the mouthparts are completely formed and the macronucleus is fully condensed. At 7 hr the membranellar band and mouth migrate to the anterior end of the cell. The macronucleus extends into a sausage-like shape. By 8 hr after sucrose shock the *Stentor* is fully regenerated. The membranellar band completely wraps around the anterior of the cell, all of the oral cilia coordinate to form a single metachronal wave, the macronucleus is re-nodulated, and the cell resumes normal feeding activity. (**B**) Surgical bisection of *Stentor*. When a *Stentor* cell is cut in half perpendicular to the long axis, two cell fragments are produced, an anterior half-cell and a posterior half-cell. Immediately after cutting, both half-cells heal their wounded edges. The anterior half-cell then regenerates a new posterior body including the hold-fast, and the posterior half-cell regenerates a new anterior body including the oral apparatus (OA). Oral regeneration in the posterior half-cell has the same general morphological events and timing as oral regeneration in the sucrose-shocked cells. Both fragments are able to regenerate because the elongated macronucleus that is divided into both halves during surgery is highly polyploid, ensuring that each half-cell retains many copies of the genome. (**C**) Comparative transcriptional profiling. We performed RNA sequencing (RNAseq) on sucrose-shocked cells regenerating as in panel A, as well as on both the anterior and posterior half-cells regenerating after bisection as in panel B. These three datasets are represented by the three circles of the Venn diagram, with blue representing genes expressed during regeneration in bottom half-cells after bisection, green representing genes expressed during regeneration in top half-cells after bisection, and coral representing genes expressed during regeneration in cells following sucrose shock. Genes were grouped into modules according to correlated expression patterns shared between two or more fragments. For example, the 'general regeneration' module was defined based on genes showing differential expression in all three cases of regeneration (OA, anterior half, and posterior half).

*1968*; *Younger et al., 1972*) and it is also known that overall levels of RNA synthesis increase several fold during the OA regeneration process (*Ellwood and Cowden, 1966*; *Burchill, 1968*; *Younger et al., 1972*).

The second regeneration paradigm we consider is bisection (*Figure 1B*). When a *Stentor* cell is cut in half along its longitudinal axis, the posterior half regenerates a new OA, while the anterior half regenerates a new tail (*Morgan, 1901b*). Thus, the list of genes expressed during regeneration in posterior halves of bisected cells presumably includes genes involved in building a new OA, but is also expected to include genes involved in building other anterior structures such as the contractile vacuole, while the list of genes expressed in the anterior half would include genes involved in rebuilding lost posterior structures. Both halves are expected to express genes involved in recovering the normal metabolic state of the cell following closure of the large wound created in the plasma membrane during surgical bisection.

Given that transcription is required for regeneration, there is likely to be a set of genes whose products drive the regenerative process. By learning the identity of these genes, we can determine the molecular pathways and building blocks involved in building new cellular structures. Proteomic analysis (*Lin et al., 2022*) has shown that the OA, anterior half-cell, and posterior half-cell, all have distinct protein compositions, hence we expect their regeneration to involve expression of different sets of genes. Transcriptional analysis provides a complementary method to proteomics for identifying new components of distinct cellular structures in *Stentor* while also allowing identification of genes whose products are needed to build the structure but that do not encode components of the final structure, and so would be missed by proteomics. This type of approach has previously been used successfully to identify genes involved in ciliogenesis, by identifying genes expressed in cells as they regenerate flagella (*Schloss et al., 1984*; *Stolc et al., 2005*; *Albee et al., 2013*). In addition to producing lists of candidate genes involved in assembly, the timing with which different genes are expressed will potentially reveal sequential steps in the assembly process itself. Furthermore, by revealing the timing of gene expression during regeneration, transcriptional analysis can provide clues about how the process is initiated. OA regeneration can be triggered by several different procedures, including removal of the OA but also rotation of the OA relative to the rest of the cortical pattern (*Tartar, 1961*) raising the question of how the cell recognizes these geometric perturbations. One approach is to determine which genes are expressed earliest in the pathway, and then move upstream to identify signals required to turn these early-expressed genes on. In terms of the timing and sequential logic of the expression program itself, we note that even if the ultimate goal is to understand how *Stentor* achieves complex spatial patterning, pattern formation is an inherently spatiotemporal process, such that both spatial and temporal control are important.

It has been possible to use RNAseq to show that a portion of the genome becomes expressed during regeneration in *Stentor* (*Sood et al., 2017*; *Onsbring et al., 2018*; *Wei et al., 2020*). These studies focused on two different regeneration paradigms. In *Sood et al., 2017*, and *Wei et al., 2020*,

genes were identified that were upregulated during OA regeneration in *S. coeruleus*, induced using sucrose shock or urea shock, respectively. The list of genes identified was presumably a mixture of those involved in building the new OA and genes involved in recovery from the stress of sucrose or urea shock that was used to remove the OA. In *Onsbring et al., 2018*, regeneration of bisected *Stentor polymorphus* cells were analyzed. Each of these three prior studies focused on a different regeneration paradigm in different species. However, we hypothesized that by comparing the transcriptional programs of regeneration in sucrose-shocked versus bisected cells, all within the same species, it would become possible to distinguish shared modules of gene expression, common to all regenerative processes, from structure-specific regeneration modules.

Here, we report a comparative transcriptomic analysis in which RNAseq was used to analyze gene expression during OA regeneration following sucrose shock as well as regeneration in posterior and anterior cell fragments following surgical bisection. By comparing gene sets expressed in these different situations, we can identify genes specific to regeneration of the OA and the posterior tail, as well as additional sets of genes involved in the recovery from surgical wounding. The timing with which specific groups of genes are expressed correlates with the formation of specific cellular structures. These results suggest a modular organization of the regeneration program. By focusing on earlier stages of regeneration, we identified conserved transcriptional regulators as well as RNA-binding proteins that are differentially expressed during regeneration. Inhibition of protein translation allows the transcriptional program to be divided into an early set of genes whose expression does not require protein synthesis, and a later set of genes whose expression is apparently dependent on production of proteins encoded by the early genes, suggesting a cascade-like logic. This work opens a new window into the molecular details underlying the century-old question of regeneration in this extraordinary single-celled organism.

## Results

### Identifying gene expression modules by comparative transcriptomics

*S. coeruleus* cells were subjected to sucrose shock to remove the OA (*Figure 1A*), or else surgically bisected to produce anterior and posterior half-cells (*Figure 1B*). These experiments produced three regenerating samples: intact cells from which the OA had been removed, which then regenerated the OA; posterior half-cells which regenerated a new anterior half, including the OA and other anterior structures; and anterior half-cells which retained the pre-existing OA and regenerated a new posterior half including contractile tail and holdfast.

For each sample, cells were collected at regular intervals and analyzed by RNAseq (see Materials and methods). The process of building a new OA takes ~8 hr as detailed in *Figure 1*. RNA samples from ~20 cells were collected prior to sucrose shock, at 30 min and at 1, 1.5, 2, 3, 4, 5, 6, and 7 hr after sucrose shock. The number of time points was based on the time required to complete regeneration. At each stage, RNA was extracted, and RNAseq libraries were sequenced (see Materials and methods). We combined reads from all samples and replicates and used these to assemble a genome-guided de novo transcriptome (*Grabherr et al., 2011*) using TRINITY. To identify genes with dynamic expression patterns during regeneration, we first mapped reads from each sample to this transcriptome and then identified differentially expressed genes over the time course in each sample. To identify genes that were differentially expressed we compared two models – a generalized additive model where changes in expression over time are modeled by natural splines, and one in which there is no dependence on time.

Overall, we identified 4323 genes that exhibited dynamic expression patterns through regeneration in sucrose-shocked cells, 1020 in posterior half-cells following bisection, and 229 in anterior half-cells following bisection. As indicated in *Figure 1C*, these three samples showed partially overlapping expression patterns, but each also expressed its own unique set of genes. The anterior half-cells showed by far the smallest number of differentially expressed genes, while the sucrose-shocked cells showed the largest. In total we detected 4811 differentially expressed genes, which constitutes roughly 10% of the *Stentor* genome. Based on the Venn diagram in *Figure 1C*, we defined six sets of differentially expressed genes: genes expressed in both sucrose shock and regenerating posterior halves, which we take to indicate genes required for OA regeneration; sucrose shock-specific, which we interpret as reflecting aspects of OA regeneration in sucrose-shocked but not bisected cells,

possibly including osmotic response to the sucrose shock itself; genes expressed only in regenerating posterior halves, which we interpret as relating to regeneration of anterior structures other than the OA; genes expressed only in regenerating anterior halves, which we interpret as being involved in regenerating the tail of the cell; genes expressed in both regenerating half-cells but not sucrose shock, which we term bisection-specific; and, finally, genes expressed in all three samples, which we take to indicate general regeneration genes. Each of these sets of genes were clustered using clara (*Kaufman and Rousseeuw, 1990*).

## Differentially expressed genes specific for OA regeneration

OA regeneration can be stimulated in a number of ways. If *Stentor* cells are treated with sucrose, the OA detaches via an autotomy process (*Tartar, 1957*). If *Stentor* cells are cut in half, the posterior half will regenerate a new OA. Analyzing genes expressed in either situation alone will reveal OA regeneration-specific genes but also genes that may be induced by the stresses of sucrose shocking or bisection, respectively. Thus, in order to obtain a list of highest confidence OA-specific genes, we compared sucrose-shocked cells and regenerating posterior half-cells in order to identify a set of genes that are differentially expressed in both samples and thus likely to be specific for OA regeneration. These genes fall into five clusters (*Figure 2A*). Cluster 1 consists of genes whose expression starts out high at t=0 (prior to removal of the OA) and decreases during the course of regeneration, possibly suggesting that these genes encode proteins whose function is dispensable for, or possibly even inhibitory of, the regeneration process. The other four clusters correspond to upregulated genes whose peak of expression takes place at successively later times during regeneration. The identity of OA-specific genes corresponding to gene models in the published *Stentor* genome (StentorDB) are listed in *Supplementary file 1*. Classes of genes present in the OA-specific module are categorized in *Figure 2B*,which includes genes encoding centriole-related proteins, kinases, and RNA-binding proteins.

Among the annotated genes in the OA-specific set (*Supplementary file 1*), many were found that are related to centriole biogenesis, in keeping with the fact that a large number of new basal bodies form during OA regeneration in order to act as basal bodies for the cilia of the MB of the OA. *Table 1* lists the key centriole-related genes found to be upregulated in the OA-specific module. It is notable that most of these are expressed in clusters 2 and 3, consisting of genes whose expression peaks at 1.5–2 hr after sucrose shock, during the period of time at which the first evidence of an oral primordium becomes visible in scanning electron microscopy (*Paulin and Bussey, 1971*), and at which transmission electron microscopy reveals that thousands of new basal bodies are being formed de novo, creating a so-called 'anarchic field' (*Bernard and Bohatier, 1981*). These basal bodies will ultimately organize themselves into arrays and become the basal bodies that nucleate the ciliature of the OA. Among the upregulated genes are SAS6, SAS4, POC1,CETN3, CETN2, CEP76, CEP135, CEP120. SAS6 is notable as one of the earliest known factors involved in assembling the ninefold symmetric structure of the centriole (*Leidel et al., 2005*). Out of the 29 most conserved ancestral centriole genes, we find that 10 are expressed in either cluster 2 or 3. This expression of the core centriole gene set at the exact stage when basal bodies are forming thus provides a biological confirmation of our analysis. One centriole-related gene, LRC45, is expressed later than the others in cluster 4. Implications of this delayed expression of a centriole-related gene will be discussed below. A handful of genes involved in ciliary assembly are expressed in later clusters, including several IFT proteins, but almost no genes encoding components of the motile ciliary machinery were differentially expressed in any cluster.

## Sucrose shock-specific genes

In contrast to the OA-specific module, which was defined by looking for genes shared in common between the expression program during OA regeneration in bisected cells and sucrose-shocked cells, the sucrose shock-specific module groups genes differentially expressed only in sucrose-shocked cells. *Figure 2C* depicts genes showing differential expression in sucrose-shocked cells but not in regenerating posterior or anterior half-cells. These genes may be involved in stress response to the sucrose shock, or remodeling the remnant of the previous OA left behind after the shock. They may also represent redundant additional OA biogenesis factors that happen to be expressed during sucrose shock only.

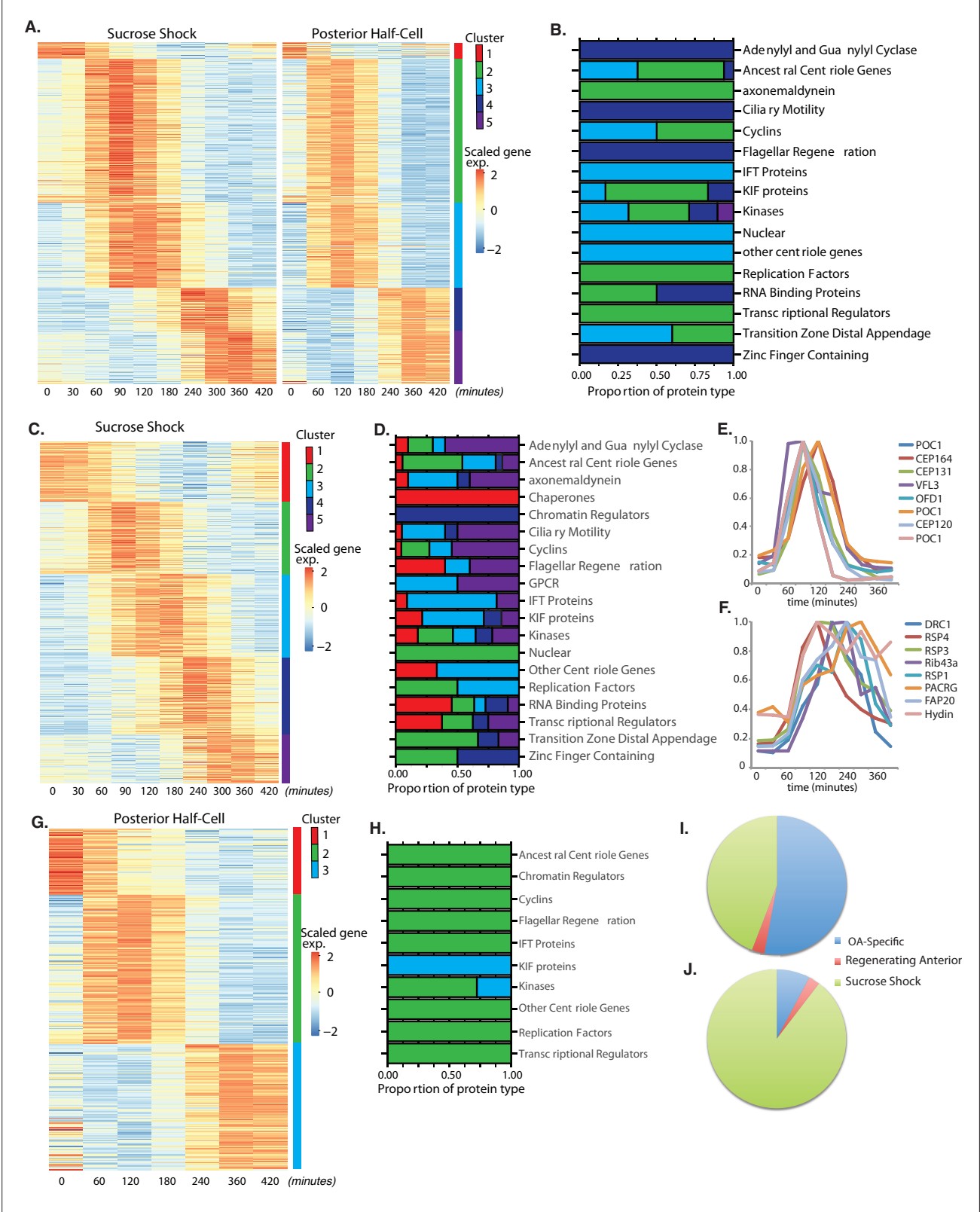

**Figure 2.** Oral apparatus (OA) regeneration program.

(**A**) OA-specific gene expression profile determined using combined sucrose shock and bisection samples. Genes are clustered into five groups using 'clara' clustering (as indicated by the colored bar on the y axis). Time since sucrose shock or bisection (in minutes) is on the x-axis. Group 1 contains all genes whose expression decreases during regeneration compared to initial levels. The peak expression of each cluster of genes corresponds with

*Figure 2 continued on next page*

*Figure 2 continued*

major developmental features identified morphologically (as described in *Figure 1A*). Z-score is calculated per row and color coded with shades of red representing expression higher than the row average and blue lower than the row average. (**B**) Proportion of gene types in each cluster for a set of reference gene classes. (**C**) Expression heatmap of genes expressed in sucrose-shocked cells that are not included in the OA-specific set of panel A. (**D**) Proportion of gene types in each cluster for sucrose shock-specific genes. (**E**) Expression time course of a selected set of canonical centriole-related genes, showing their coordinate expression at a time in regeneration when numerous centriole/basal bodies are forming at the start of OA biogenesis. (**F**) Expression time course of a set of highly conserved genes expressing proteins known to be involved in ciliary structure and motility. Comparison with panel E illustrates that cilia-related gene expression occurs later than centriole-related gene expression. (**G**) Expression heatmap of gene expressed in regenerating posterior half-cells that are regenerating new anterior portions, showing genes specific to the regenerating posterior and not included in the OA-specific set of panel A. (**H**) Proportion of gene types in each cluster for regenerating posterior half-cells. (**I,J**) Pie charts showing fraction of all centriole (**I**) and cilia (**J**) genes showing upregulation in the two paradigms (sucrose shock and posterior halves), illustrating that most upregulated cilia-related genes are specific to sucrose shock.

Similar to the OA-specific module (*Figure 2A*), we observe five clusters of gene expression among the sucrose shock-specific genes (*Figure 2C*). The timing of these clusters matches that seen in the OA-specific module. The identity of sucrose shock-specific genes corresponding to gene models in the published *Stentor* genome (StentorDB) are listed in *Supplementary file 2*. Genes upregulated during sucrose shock response included 12 of the 29 ancestral centriole genes. As with the OA-specific gene set, these genes were enriched in cluster 2 of the sucrose shock response, for example POC1, OFD1, CEP164, and CP131/AZI. In many cases, the sucrose shock module includes paralogs

**Table 1.** List of centriole/basal body genes in the oral apparatus (OA)-specific module (*Figure 2A*).

| | Cluster of peak expression | | | | |
|---|---|---|---|---|---|
| | 1 | 2 | 3 | 4 | 5 |
| SAS6 | | + | | | |
| STIL | | + | | | |
| POC5 | | + | | | |
| POC11/CCD77 | | + | | | |
| POC18/WDR67/Tbc31 | | + | | | |
| VFL3 | | + | | | |
| GCP2 | | + | | | |
| GCP4 | | + | | | |
| Jouberin | | + | | | |
| MKS1 | | + | | | |
| MKS6 | | + | | | |
| XRP2/TBCC | | + | | | |
| RTTN | | + | | | |
| CCD61 | | + | | | |
| Bld10/CEP135 | | + | + | | |
| CEP350 | | + | + | | |
| POC16 | | + | + | | |
| Centrin 2 | | + | + | | |
| CEP44 | | | + | | |
| POC12 | | | + | | |
| Centrin 3 | | | + | | |
| MKS3 | | | + | | |
| LRRC45 | | | | + | |

of these genes that were not expressed in the OA-specific module, consistent with the idea of redundancy between bisection and sucrose shock-induced OA formation. Cluster 3 contains genes relating to ciliary assembly such as the intraflagellar transport proteins. Within the sucrose-specific module, the expressed paralog of the key centriole biogenesis initiating protein SAS6 is not expressed highly until late in regeneration (cluster 5). The expression timing of a representative set of centriole-related genes is given in *Figure 2E*.

In stark contrast to the lack of cilia-related genes in the OA-specific gene set presented above, we found that clusters 3 and 4 of the sucrose shock-specific gene set contain a large number of genes that encode proteins components of motile cilia. Expression of genes encoding components of the inner and outer dynein arms, which power ciliary motility, are seen across clusters 2, 3, and 4. In addition to the dynein arms themselves, two other multi-protein complexes are required to coordinate dynein activity, radial spokes (*Smith and Yang, 2004*), and the dynein-regulatory complex (*Viswanadha et al., 2017*). Genes encoding the dynein regulatory complex are expressed exclusively during cluster 3 (DRC 1, 2, 3, 4, 5, 7, 9, 10, and 11). Expression of genes encoding radial spoke components begins in cluster 3 and is most apparent in cluster 4, where eight radial spoke proteins are expressed (RSP1, 3, 4, 7, 9, 10, 14, and 16). The radial spokes interact with the central pair microtubule complex, and several central pair-specific proteins are upregulated during clusters 2–4 (PF6, PF20, CPC1, Hydin). The axoneme of motile cilia contains structural proteins that are located at the junction between the A and B tubules. These junctional proteins are expressed in clusters 3 and 4 (Rib43a, Rib72, PACRG, FAP20/BUG22). We thus find that clusters 3 and 4, ranging in expression timing from 120 to 300 min, contain many genes involved in supporting ciliary motility. None of these genes is required for the assembly of cilia, but instead are involved in coordinating the activity of axonemal dyneins to generate motility (*Zhu et al., 2017*). In *Chlamydomonas*, radial spoke protein synthesis reaches its maximum rate 30–60 min after the flagella have begun assembling onto pre-existing basal bodies (*Remillard and Witman, 1982*). This timing roughly matches the delay of 1 hr seen in our data between the peak expression of genes involved in ciliary assembly (cluster 2) and genes encoding radial spokes during *Stentor* regeneration (cluster 4). The timing of group 4 also correlates with the time period during which the oral cilia transition from their initial random beating motility to their characteristic coordinated beating motility, forming metachronal waves (*Paulin and Bussey, 1971*; *Wan et al., 2020*). The expression timing of a representative set of genes encoding protein components of motile cilia is given in *Figure 2F*.

Overall, we observe that the peak expression of genes encoding centriole proteins occurs earlier than the peak expression of genes encoding motile cilia proteins (compare *Figure 2E and F*), which is consistent with the fact that ciliogenesis takes place later than basal body biogenesis in *Stentor*.

## Regeneration of non-oral anterior structures

In bisected cells, the posterior half-cell regenerates anterior structures, including the OA but also other structures such as the contractile vacuole and the cellular anus (cytopyge). The contractile vacuole can still regenerate in enucleated posterior half-cells (*Stevens, 1903*; *Tartar, 1956*), but this does not necessarily mean that contractile vacuole-related genes are not upregulated during normal regeneration. Our data show three clusters of gene expression specific for regeneration of anterior structures in posterior half-cells (*Figure 2G and H*). Cluster 1 represents downregulated genes whose expression is highest at t=0, prior to bisection, and then decreases during regeneration. Cluster 2 spans the 60–180 min time points and therefore matches the expression of cluster 3 of the OA-specific program (*Figure 2A*). In cluster 3, expression drops midway through regeneration (120 min) and then rises and peaks around 360 min (240–420 min). This expression thus resembles the timing of cluster 5 of the OA-specific gene set. Not only does the timing resemble cluster 5 of the OA-specific genes, the types of genes that are in this cluster are also similar, specifically EF hand, shippo-rpt proteins, and glutathione *S*-transferase. The identity of anterior regeneration-specific genes corresponding to gene models in the published Stentor genome (StentorDB) are listed in *Supplementary file 3*.

The same lack of cilia-specific gene expression that is seen in the shared OA regeneration (*Figure 2A*) is also seen in posterior halves regenerating anterior structures. If we consider all clearly annotated genes encoding centriole structural proteins or assembly factors (*Figure 2I*), including those discussed above, we see that the vast majority were observed in the sucrose shock experiment.

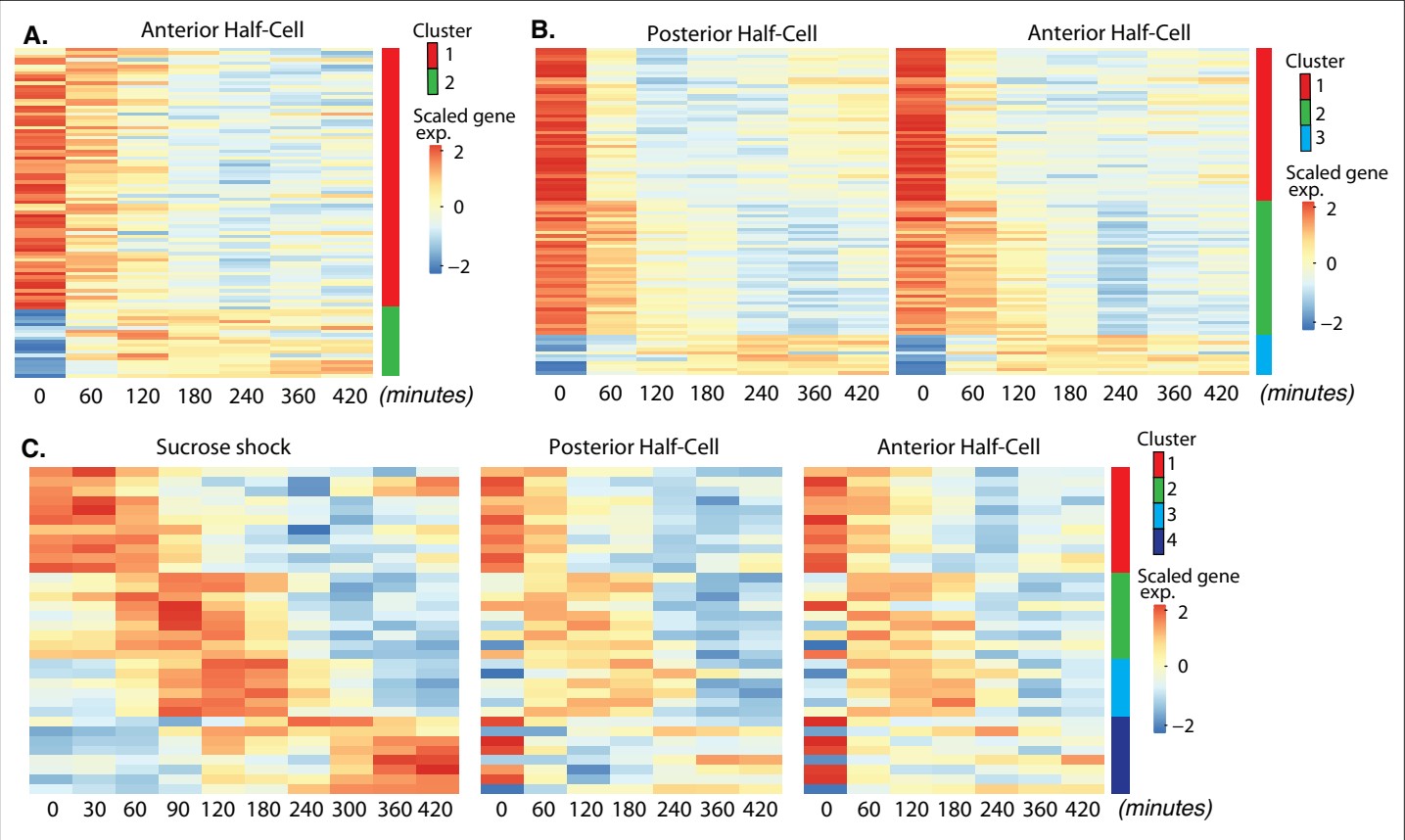

**Figure 3.** Expression modules distinct from oral apparatus (OA) regeneration.
(A) Expression in anterior half-cells that are regenerating posterior tails including holdfast. (B) Genes showing correlated differential expression in both halves of bisected cells. (C) Genes showing differential expression in all three types of regenerating cell fragments, sucrose shocked, anterior half-cell, and posterior half-cell.

The same trend is seen to an even greater extent for genes encoding protein components of motile cilia (*Figure 2J*), including the examples listed above.

## Posterior tail regeneration

In a bisected cell, the anterior half-cell regenerates a new posterior half, including the contractile tail and holdfast. We refer to the gene expression module seen in these cells as the 'tail regeneration' program (*Figure 3A*). Two clusters of expression were found during tail regeneration, a cluster of genes that are downregulated during tail regeneration, and a single cluster of genes that are upregulated during regeneration. The tail regeneration-specific transcriptional program is dramatically different from that seen in OA regeneration, in terms of both the number and type of genes that show differential expression. The number of genes upregulated during tail regeneration (13) is far smaller than the number upregulated during OA regeneration (564; *Figure 2A* clusters 2–5), consistent with the fact that the posterior holdfast can still regenerate in enucleated cells and therefore does not rely on transcription for its regeneration (*Tartar, 1956*). Increased expression for many of the tail regeneration genes does not occur until at least 120 min, by which time it has been reported that the holdfast has already completely regenerated (*Morgan, 1901a*; *Weisz, 1951*), suggesting that the upregulation of these genes is taking place after assembly, perhaps to regenerate depleted pools of precursor protein.

Unlike OA regeneration, tail regeneration does not involve expression of any known centriole or cilia-related genes. Instead, the expressed gene set is dominated by EF hand proteins, which constitute four of the nine upregulated genes for which annotation data exists. Studies of cellular structure in *Stentor* showed that the posterior half of the cell contains long contractile fibers composed of centrin-like EF hand proteins (*Huang, 1973*; *Maloney et al., 2005*), and studies of cell movement

showed that the contractile behavior of the cell, which is driven by these EF hand protein fibers, occurs primarily in the posterior half of the cell (*Newman, 1972*). The identity of tail regeneration-specific genes corresponding to gene models in the published *Stentor* genome (StentorDB) are listed in *Supplementary file 4*.

## Bisection-specific genes

Sucrose shock removes the OA cleanly, without creating a wound. In contrast, surgical bisection disrupts the membrane with visible loss of cytoplasm. In order to investigate the molecular response to this wounding, we looked for genes that showed similar patterns of differential expression in both halves of bisected cells, but not in sucrose-shocked cells. This analysis revealed 98 differentially expressed genes which grouped into three clusters based on temporal pattern (*Figure 3B*). The identity of bisection-specific genes corresponding to gene models in the published *Stento*r genome (StentorDB) are listed in *Supplementary file 5*.

Cluster 1 consists of genes that are turned off rapidly during regeneration of bisected cells, and consists predominantly of metabolic enzymes and chaperones. Cluster 2 are genes that turn off more gradually during bisected regeneration, with expression levels reducing during the first hour of regeneration. This cluster includes a number of proteases. We noted that cluster 2 of the bisection response (slow downregulation) contains a number of genes encoding protein classes similar to those seen in cluster 1 (rapid downregulation) of the OA-specific gene set. Specifically, both gene sets include orthologs of von Willebrand factor domain protein, serine carboxypeptidase, papain family cysteine protease, glycosyl hydrolase, and aldo/keto reductase. The genes are different in the two datasets but encoding similar proteins. This similarity suggests that similar genes are inactivated during regeneration in bisected and sucrose-shocked cells, but with slower kinetics of repression in the bisected cells.

Cluster 3 contains genes whose expression increases during bisected regeneration, with the peak of expression generally in the range of 120–240 min. This cluster includes membrane transporters as well as carbonic anhydrase.

## Genes shared by all regeneration processes

Is regeneration a single process, or a collection of distinct processes that depend on which part is missing? By considering the overlap of expression patterns among all samples analyzed, we identified candidates for general regeneration genes expressed during all forms of regeneration (*Figure 3C*). The identity of the general regeneration genes corresponding to gene models in the published *Stentor* genome (StentorDB) are listed in *Supplementary file 6*. Far fewer genes were contained in this group than in any other regeneration module, suggesting that most regeneration genes are specific to distinct aspects of regeneration. The small number of general genes could potentially play a role in building replacement cortical structures shared by all parts of the cell or in recognition by the cell that regeneration is taking place.

Cluster 1 consists of genes whose expression is reduced in all forms of regeneration. This cluster consists mostly of genes encoding metabolic proteins, and suggests a general trend, also seen in other modules, for the cell to downregulate metabolic activity during regeneration. Clusters 2 and 3 are genes upregulated in all forms of regeneration, while cluster 4 consists of genes that show differential expression in all three cases, but unlike clusters 2 and 3, cluster 4 genes are upregulated following sucrose shock but downregulated in both halves of bisected cells. The genes in these clusters do not fall into any discernable functional families.

## Regeneration in the absence of translation

A central question in regeneration is the nature of the stimulus that triggers the appropriate transcriptional response. As an initial step toward addressing that question, we ask whether the triggering signal, whatever it is, directly drives expression of all the genes upregulated during regeneration (*Figure 4A*) or whether, instead, the triggering signal may drive a subset of genes which then, in turn, drive subsequent rounds of gene expression, thus leading to a cascade-like mechanism (*Figure 4B*). In order to distinguish between these direct and cascade schemes, we repeated the RNAseq analysis of OA regeneration in sucrose-shocked cells treated with cycloheximide to block translation. This treatment would not be expected to affect pre-existing proteins that constitute the triggering stimulus

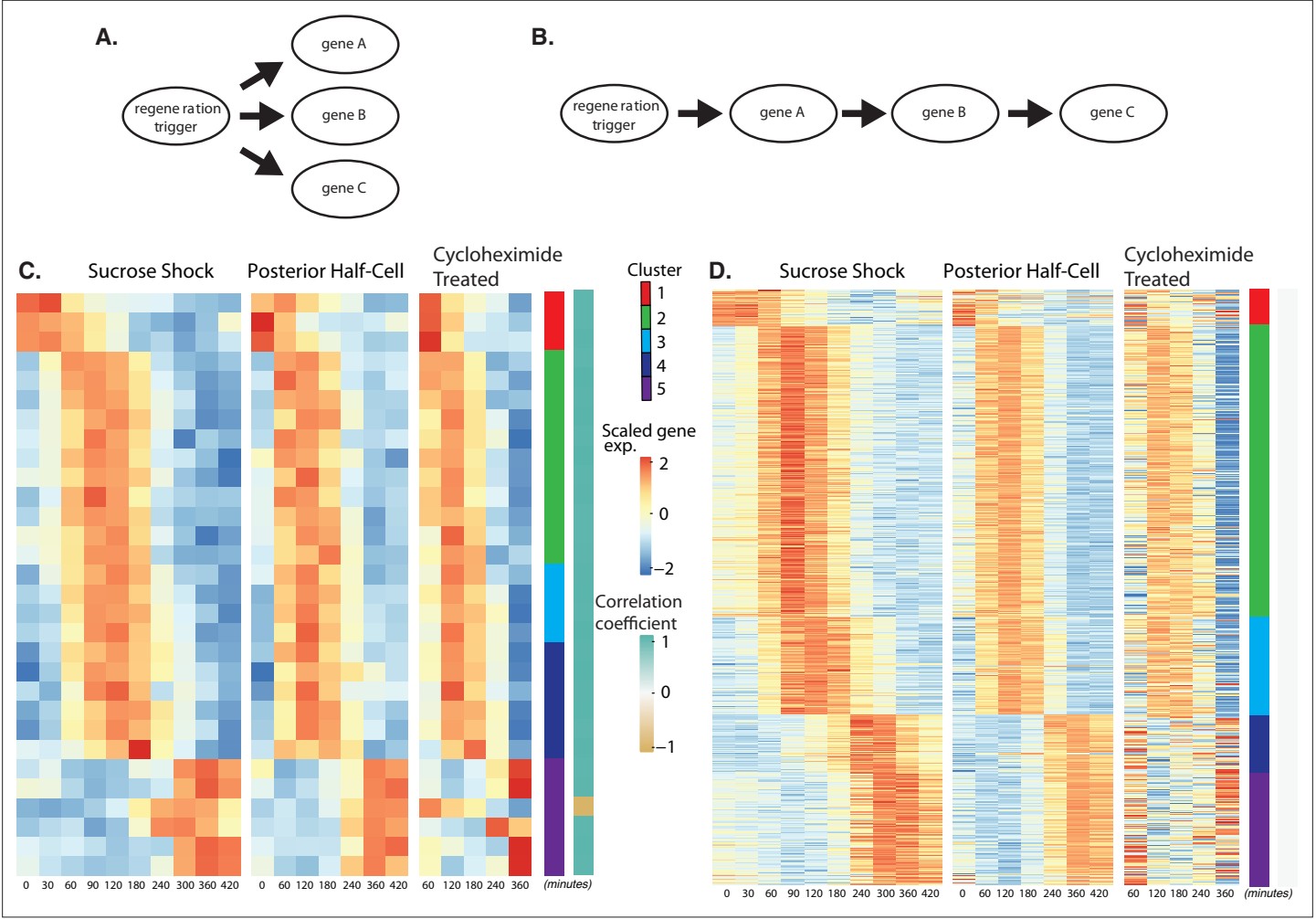

**Figure 4.** Regeneration in the absence of protein synthesis. (**A**) Direct model of coordination, in which the regeneration-triggering stimulus directly triggers each gene. (**B**) Cascade model, in which regeneration-triggering stimulus drives one gene, whose product then drives another gene, with potentially multiple steps being required to ultimately trigger all genes in the cascade. (**C**) Heatmap showing OA-specific genes whose expression pattern in cycloheximide cells is highly correlated with the expression pattern in untreated cells. (**D**) Heatmap showing OA-specific genes whose expression pattern in cycloheximide cells is uncorrelated with the expression pattern in untreated cells.

pathway, but would prevent any of the direct targets of the triggering stimulus from being produced to trigger subsequent waves of expression.

Out of the 431 genes that show differential expression in the OA-specific regeneration module (*Figure 2A*), we found that 21 genes, constituting only 5% of the total gene set, were completely unaffected by cycloheximide treatment as judged by a high correlation coefficient in expression profiles with and without cycloheximide (*Figure 4C*). The remaining 411 genes (95% of the total) had their expression affected to varying degrees as judged by a reduction in the correlation with their untreated expression pattern (*Figure 4D*). The average correlation coefficient between cycloheximide-treated and -untreated genes, for clusters 1–5, were 0.72, 1.54, 2.31,–0.22, and 0.81, respectively. Based on these average correlation coefficients, the largest effects, corresponding to the lowest correlations, were seen in cluster 1, consisting of genes that normally are repressed during regeneration, and in clusters 4 and 5, consisting of genes that normally are upregulated late in regeneration. In the case of cluster 1, there was a general loss of repression when translation was blocked. In the case of clusters 4 and 5, there was an overall reduction in upregulation when translation was blocked.

The results suggest a hypothetical cascade model in which one set of genes are directly triggered by a pathway that relies entirely on existing proteins, and then one or more of these gene products trigger the rest of the program, possibly by acting as transcription factors. Given the position of these

genes at the start a regulatory cascade (*Figure 4B*), we would expect that these genes might be expressed early in the overall program, since they would not be able to cause changes in expression that take place before they, themselves, are expressed. Consistent with this view, the majority of the genes unaffected by cycloheximide treatment are contained in cluster 2, the earliest expressing cluster, while the loss of expression in cycloheximide-treated cells is most dramatic in clusters 4 and 5, the latest-expressing clusters.

It has previously been shown that treatment of regenerating *Stentor* cells with cycloheximide leads to arrest of regeneration if treated within the first 4–5 hr, whereas when cycloheximide is added at later times it causes a delay in the final stages but does not fully prevent regeneration (*Burchill, 1968*; *Younger et al., 1972*). We note that the time range in which cycloheximide treatment switches from causing arrest to causing a delay is approximately the time at which we see cluster 4 showing its increased gene expression.

## Transcriptional regulators expressed during regeneration

The cascade-like organization of regeneration described in *Figure 4* suggests that transcription factors expressed early in the regeneration program (cluster 2) might trigger genes at subsequent stages of the process. A search for transcription factors in the OA-specific cluster 2, which consists of the earliest genes to be upregulated during OA regeneration, revealed the transcription factor E2F, as well as its dimerization partner E2FDP1, and several alleles of Rb, a regulator of the E2F-DP1 interaction. The Rb-E2F-DP1 module plays important roles in regulating cell cycle-dependent processes in many species (*Bertoli et al., 2013*; *Nair et al., 2009*), including *Tetrahymena* and *Chlamydomonas* (*Zhang et al., 2018*. *Cross, 2020*). Because we found all three members of the Rb-E2F-DP1 module in cluster 2 of the OA-specific program, E2F is a promising candidate for a regulatory factor controlling later events in the program.

The temporal pattern of E2F expression is shown in *Figure 5A*, which indicates that E2F is expressed both in sucrose shock responding cells and posterior halves that are regenerating anterior halves, but not in anterior half-cells that are regenerating posterior tails. Importantly, E2F is upregulated in OA-regenerating cells even if they are treated with cycloheximide, consistent with a possible role as an early gene that may serve as a regulator of later expression waves. To explore this possibility, we identified putative E2F targets based on promoter motif analysis (*Rabinovich et al., 2008*), and asked whether these predicted targets exhibited specific expression patterns during regeneration. As shown in *Figure 5B*, there is indeed a tight pattern of E2F targets expressed during a 1 hr window that corresponds roughly to the time in regeneration at which centriole-related genes are expressed (*Figure 2E*). This pattern closely matches the pattern of expression of the E2F ortholog, designated E2F-1. The identities of predicted E2F targets among the OA-specific genes are annotated in *Supplementary file 1*. Based on the data annotated in this table, there are 13 predicted E2F targets, of which only one, SteCoe_2152, showed normal expression when cells were treated with cycloheximide. The other 12/13 of these targets show reduced expression compared to untreated cells. The majority (10/13) of the E2F targets are in cluster 2, two are in cluster 3, and one is in cluster 5. None of the predicted E2F targets are found in cluster 1 consisting of genes that are repressed during regeneration, suggesting that E2F likely plays an activating function in this process. The predicted E2F targets do not fall into any single characteristic functional families. However, consistent with the role of E2F in regulating cyclin transcription in other organisms, the target list in *Stentor* does include cyclin and cyclin-associated protein A. In contrast to the identification of multiple predicted E2F target genes among the genes upregulated during regeneration after either sucrose shock or in posterior half-cells that are regenerating anterior structures, predicted E2F targets were not observed among genes upregulated during tail regeneration in anterior half-cells.

We tested whether E2F was functionally important for OA regeneration using RNAi directed against the upregulated E2F ortholog SteCoe_12750 (*Figure 5C–K*). In E2F RNAi-treated cells, the oral primordium formed normally after sucrose shock (compare *Figure 5C* versus *Figure 5G*), but failed to progress or be maintained, such that by 8 hr, a large fraction of cells were missing either an oral primordium or fully developed OA (*Figure 5E* versus *Figure 5I*). Even after 24 hr, many RNAi cells lacked any sign of OA or primordium. Taken together, our results indicate that oral regeneration proceeds normally for the first 4 hr, but then fails to proceed in later time points (*Figure 5K*). The loss

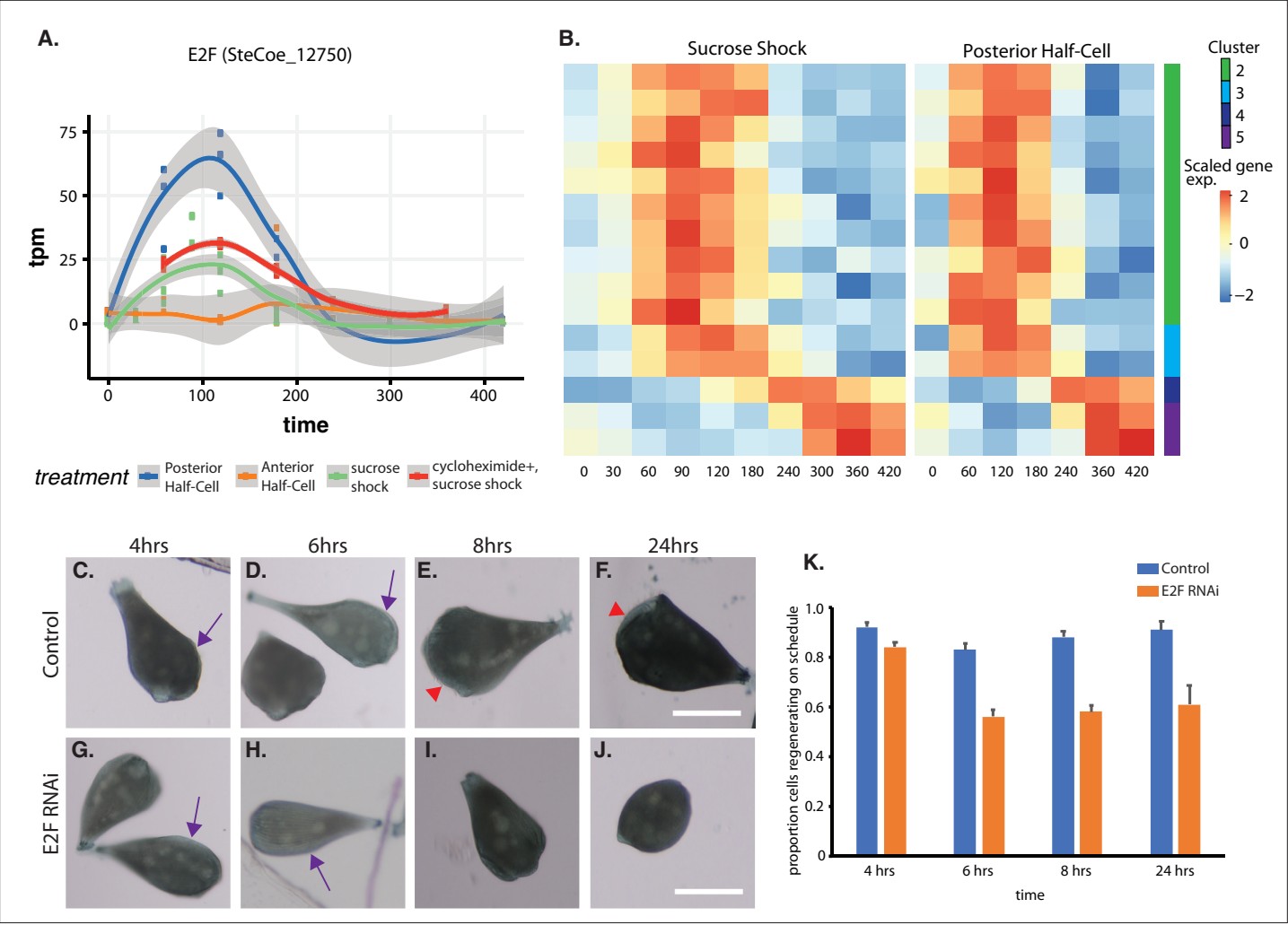

**Figure 5.** Expression of E2F and its targets during oral apparatus (OA) regeneration. (**A**) E2F is an early-expressed gene. Plot shows expression of E2F homolog SteCoe_12750 as a function of time including in cycloheximide-treated cells. OA (blue curve) indicates expression in sucrose-shocked cells. The purple curve shows that E2F is still upregulated in cells in which protein synthesis is inhibited. (**B**) Expression heatmap of predicted E2F targets in the OA-specific expression module. Clusters refer to clusters in the original clustering of OA-specific genes (**Figure 2A**). Cluster 1 is not included in the plot because there were no predicted E2F targets among the cluster 1 genes. (**C–F**) Examples of control cells at 4, 6, 8, and 24 hr after regeneration. Purple arrows mark the location of oral primordium. Red arrow head marks the location of a fully formed OA. (**G–J**) Examples of E2F RNAi cells at 4, 6, 8, and 24 hr, illustrating the proper formation of the oral primordium by 4 hr, but lack of visible primordium or OA at later stages. All scale bars 200 μm. (**K**) Fraction of cells regenerating properly at each time point. For the 4, 6, and 8 hr time points, on-schedule regeneration is defined as cells having a visible oral primordium (4 hr), a full length primordium extended along the side of the cell (6 hr), and an oral apparatus located at the anterior end of the cell whose diameter is at least 50% of the diameter of the cell (8 and 24 hr). Total number of cells analyzed from five separate biological experiments was n=206 for control and 310 for RNAi. Error bars represent standard error for the proportions expected based on a Bernoulli distribution. For all four time points, the proportion of RNAi cells regenerating on schedule (defined according to a set of milestone events as described in Materials and methods) was significantly less than for controls, with p values of (p=0.007, p<0.0001, p<0.0001, and p=0.0005) respectively based on Fisher's exact test.

The online version of this article includes the following figure supplement(s) for figure 5:

**Figure supplement 1.** Additional information for **Figure 5**.

of oral primordia at later stages is reminiscent of the loss of oral primordia in regenerating cells from which the nucleus is removed (**Tartar, 1961**).

## A Pumilio ortholog and its targets expressed during regeneration

An important question is to what extent do patterning mechanisms in *Stentor* involve similar molecules or pathways compared to developmental mechanisms in animals, particularly developmental mechanisms that may occur at the one-cell stage. One of the most highly conserved developmental

regulators in early animal development is the RNA-binding protein Pumilio, which mediates mRNA localization and translation control during pattern formation in animal embryos such as *Drosophila* (*Wreden et al., 1997*; *Gamberi et al., 2002*; *Sonoda and Wharton, 1999*). Our differential expression analysis identified five Pumilio orthologs showing differential gene expression during regeneration, one specific to OA regeneration (cluster 2; SteCoe_27339; *Figure 6A*), one specific to tail regeneration (cluster 2; SteCoe_37495; *Figure 6B*), and three specific to the sucrose shock response (cluster 1; SteCoe_9692; cluster 2 SteCoe_16166; cluster 5; SteCoe_22534). The tail-specific Pumilio SteCoe_37495 also shows differential expression during OA regeneration, but with different timing (*Figure 6B*). These putative Pumilio orthologs all contain PUF domains based on Pfam analysis, and phylogenetic analysis groups them together with bona fide Pumilio orthologs in animals (*Figure 6C*). No Pumilio orthologs were identified as having differential expression in the anterior regeneration, bisection-specific, or general regeneration datasets.

We hypothesized that these Pumilio orthologs may play a role in regulating the localization or translation of other regeneration-specific messages. If this were true, then we would expect the gene expression program of regeneration to include genes whose messages contain Pumilio-binding sites. Analysis of Pumilio recognition motifs (*Ray et al., 2013*; see Materials and methods) among the set of differentially expressed genes showed that indeed there were differentially expressed genes that contained predicted Pumilio recognition sites, and that these were located in the 3' UTR region (*Figure 6D*). Among the OA regeneration specific genes, 34 genes were found to be putative Pumilio targets (see *Supplementary file 1*), and these distinctly cluster into two groups with peaks of expression at 120 and 300 min respectively after the start of regeneration (*Figure 6E*). The timing of these peaks coincides with the peak expression of Pumilio orthologs. For example, the OA-specific Pumilio SteCo_27339 shows peak expression at 120 min, while SteCoe_37495 peaks at 120 min during tail regeneration but 300 min during OA regeneration. Notably, out of the 34 Pumilio target genes showing differential expression, none of them were contained in cluster 1 of the OA regeneration gene set, the cluster that contains genes whose expression decreases during regeneration. Thus, 100%of the Pumilio targets showed increased rather than decreased expression during OA regeneration. All of the predicted Pumilio targets showed reduced induction in cycloheximide-treated cells. There was no overlap between the set of predicted Pumilio targets and the set of predicted E2F targets.

Sixteen of the 34 predicted targets showed recognizable homology, and out of these, six were either kinases or phosphatases, which we speculate may potentially suggest a role for Pumilio in regulating mRNA encoding proteins of signaling pathways that regulate later steps of morphogenesis. Another four targets corresponded to basal body-associated proteins.

We tested the functional role of the OA-induced Pumilio gene SteCoe_27339 using RNAi (*Figure 6F–J*). We observed that RNAi of this Pumilio ortholog had no effect on cell morphology in pre-shocked cells (*Figure 6H*) but caused a number of morphological abnormalities when cells regenerated after sucrose shock. The most common defect seen was that cells regenerated an abnormally small OA that was significantly smaller than the diameter of the cell (*Figure 6I*). Another defect seen in some cells was a failure to build an OA at all, such that some cells had a primordium located on the side of the cell that failed to migrate to the anterior end (*Figure 6—figure supplement 1A*). Other, less frequent phenotypes were the presence of two posterior tails (*Figure 6—figure supplement 1B*), cells having an abnormal rounded appearance rather than the usual elongate Stentor cell shape (*Figure 6—figure supplement 1C*), and dead cells showing a rupture of the cell and spilling of cytoplasm (*Figure 6—figure supplement 1D*). The frequency of these phenotypes and other less frequent morphological defects are tabulated in *Figure 6J* which plots the aggregate result of three separate experiments done on different days for a total of 105 cells. The result of the individual experiments is given in *Figure 6—figure supplement 1* panels E–G, which shows that the same set of phenotypes were seen across multiple experiments.

## Discussion
### Modularity in the regeneration program

A classical question in regenerative biology is whether regeneration represents a single process that restores proper morphology following the loss of any part, or instead a collection of distinct, part-specific processes. Our comparative transcriptomic approach in *Stentor* clearly supports the latter

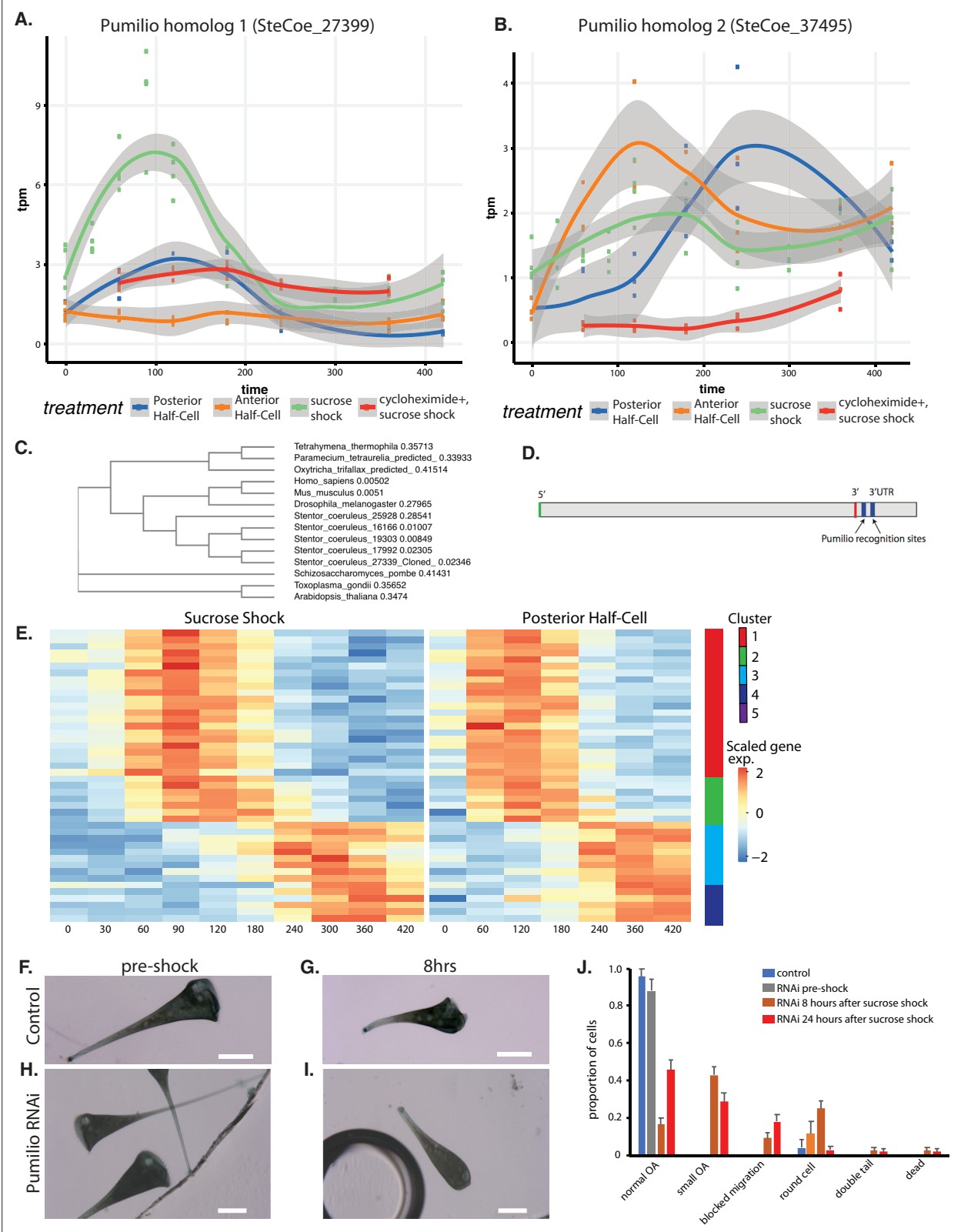

**Figure 6.** Pumilio is an early-expressed gene in oral apparatus (OA) regeneration. (**A**) Expression versus time plot of Pumilio ortholog SteCoe_27339 upregulated in both cases of OA regeneration: sucrose shock (blue curve) and regeneration of the anterior half of a posterior half-cell (red curve). Expression is still seen in cycloheximide-treated cells, indicating that this Pumilio ortholog is an 'early' gene in the cascade. SteCoe_27339 is not expressed in anterior half-cells which are not regenerating an OA. (**B**) Expression versus time of Pumilio ortholog SteCoe_37495 which is upregulated

*Figure 6 continued on next page*

*Figure 6 continued*

in all three regeneration cases, but with different timing as indicated by the blue, green, and red curves. Unlike SteCoe_27339, this Pumilio ortholog is not expressed in cells treated with cycloheximide, indicating it is a 'late' gene whose expression depends on earlier gene products of the cascade. (C) Phylogenetic tree of stentor Pumilio orthologs. (D) Example of predicted Pumilio recognition sequences in a target gene (SteCoe_10652) encoding a CAMK family kinase showing OA-specific expression. (E) Expression heatmap of predicted Pumilio targets in the OA-specific module. (F–J) RNAi analysis of Pumilio function during regeneration of OA following sucrose shock in Stentor. (F) Pre-shock control cells. (G) Control cells 8 hr after sucrose shock. (H) Pre-shock Pumilio RNAi cells showing normal cell shape and OA size. (I) Pumilio RNAi cell imaged 8 hr after sucrose shock, illustrating abnormally small OA. Scale bars 200 µm. (J) Distribution of phenotypes in RNAi of Pumilio (from three separate biological experiments, n=107). Error bars represent standard error. For images illustrating these phenotypic categories, see *Figure 6—figure supplement 1* panels A–D.

The online version of this article includes the following figure supplement(s) for figure 6:

**Figure supplement 1.** Additional information for *Figure 6*.

view, in that we found regeneration genes could be decomposed into modules specific for different aspects of regeneration. Each module contains temporally distinct waves of expression. Most of the gene expression modules can be interpreted as representing genes encoding proteins required in large quantities for regenerating various cellular structures such as the OA or the posterior holdfast. However, it is important to recognize that some of the upregulated genes may be involved not in rebuilding lost structures, but in helping the cell to recover from the stress of the perturbation that was used to drive regeneration in the first place. In the case of the bisection-specific module, we interpret these genes as involved in recovery from the surgical wounding (see below). In the case of the sucrose shock response, we expect that some of the genes upregulated during this response but not during bisection may be specific for osmotic stress incurred during sucrose treatment.

Given that the different regeneration paradigms (OA regeneration, anterior regeneration, and posterior regeneration) involve the rebuilding of different structures, and are induced by different stimuli (cutting or sucrose), the function of the general regeneration module genes (*Figure 3C*) is not clear. Since all regeneration events studied here involve rebuilding cortical patterns, we may expect that some of the general regeneration genes have to do with cortical organization. Consistent with this idea, one of the genes upregulated during general regeneration is NPHP4, a ciliary disease gene whose product is involved in linking cilia with cell polarity pathways (*Yasunaga et al., 2015*). The fact that the general regeneration module contains the fewest genes out of all the modules identified suggests that regeneration does not represent a single 'master' program of expression, but rather a composite of distinct expression modules or subroutines specific for regenerating individual parts of the cell. Somehow, the cell must recognize which part is missing and trigger the appropriate module to restore that part.

## Cascade versus production schedule

All regeneration modules showed distinct temporal waves of expression. Several features of the timing of these waves match our a priori expectations about the gene expression program of OA regeneration. First, the duration of each gene expression group is roughly 1–2 hr (*Figure 2A and C*), corresponding to the length of time that regeneration is known to proceed following surgical removal of the nucleus (*Tartar, 1961*). The persistence of regeneration over this time scale likely reflects the lifetime of the mRNAs that drive each stage. This time scale also matched the period of time during which visibly distinct morphological processes occur, for example ciliogenesis initiates in different regions of the oral primordium at slightly different times, with early-stage events of ciliogenesis taking place over a roughly 1–2 hr period (*Paulin and Bussey, 1971*). Therefore, we expected that groups of related genes would show peaks of expression lasting on the order of 1–2 hr, as we observed. Second, the number and timing of the five clusters correspond with the number and timing of known morphological events, consistent with our a priori expectation that different morphological events in regeneration may be coordinated by distinct modules of genes. Finally, we observed a strong correlation between the types of genes expressed at a given stage, and the cell biological events taking place at that stage. Based on this correlation, we believe that examination of other genes with correlated expression patterns will reveal previously unknown molecular players in organelle regeneration, centriole biogenesis, and ciliogenesis.

We can imagine two general schemes by which the timing of these waves could be determined. One model is a 'production schedule', in which a master clock triggers successive waves at different

times after the initiation of the process. The alternative is a 'cascade' model in which the products of early waves of gene expression are required to trigger expression of later waves. We note that while cascades are a widespread scheme for creating temporal programs of gene expression, there are also examples of production schedule mechanisms in which a single transcription factor drives expression of different target genes as it accumulates, an effect produced by variation among promotors in their threshold of activation, such that low threshold target genes are expressed first, followed later by high threshold target genes (*Hansen and O'Shea, 2016*; *Chen et al., 2020*).

In the production schedule type of model, later waves of expression do not depend on the products of earlier genes, and would thus occur normally even in the absence of protein synthesis. The fact that cycloheximide treatment had larger effects on later expressed groups of genes (*Figure 4C and D*) is thus more consistent with a cascade mechanism. Gene cascades are well known in many different systems, such as bacterial sporulation, bacteriophage infection, and insect morphogenesis. Our results are particularly reminiscent of events seen during the response of serum-starved mammalian cells to the re-addition of serum as well as the response of mammalian cells to viral infections. In both cases, a set of co-called immediate early genes are expressed even in the absence of protein synthesis (*Lau and Nathans, 1985*), and these genes encode factors required for driving further waves of gene expression. We propose that E2F and other early-expressed genes whose expression does not require protein synthesis may be playing an analogous role in regulating *Stentor* regeneration. The organization of a regeneration response into temporally distinct modules of gene expression is similar that seen in animal regeneration (*Monaghan et al., 2007*), which controlled by cascade-like gene regulatory networks (*Smith et al., 2011*).

In addition to a genetic cascade, regeneration involves expression of a large number of signaling molecules, in particular kinases. Indeed, kinases are expressed at all stages of regeneration, consistent with the massive expansion of the kinome in *Stentor* (*Reiff and Marshall, 2017*) although specific kinase families tend to be expressed at specific stages. One abundant class of kinases observed among the upregulated genes was the dual specificity DYRK kinases, with 13 different DYRK family members expressed (5 in the OA regeneration gene set, 8 in the sucrose shock response gene set). We note, however, that the kinome of *S. coeruleus* has been predicted to contain 142 DYRK family members, making them one of the most highly expanded kinase families in the genome (*Reiff and Marshall, 2017*). Given that the *Stentor* genome contains 35,000 genes, we would expect that roughly 14 DYRK genes would be present in any randomly chosen set of 3000 genes. The regeneration program does not, therefore, show any particular enrichment for DYRK kinases.

## Comparison to other studies of *Stentor* regeneration

Three previous studies analyzed changes in gene expression during different individual forms of regeneration in *Stentor* (*Sood et al., 2017*; *Onsbring et al., 2018*; *Wei et al., 2020*). Each of these studies addressed just one regeneration paradigm – either OA regeneration (*Sood et al., 2017*; *Wei et al., 2020*) or bisection (*Onsbring et al., 2018*), and therefore did not permit the decomposition of expression patterns into gene modules as was done in the present study.

*Onsbring et al., 2018*, analyzed transcription in bisected cells of *S. polymorphus*, a different *Stentor* species that is smaller than *S. coeruleus* and contains a green algal endosymbiont. As with our study, they found that a much larger number of genes were expressed in the posterior half-cell compared to the anterior. GO-term analysis of the *S. polymorphus* data showed enrichment for several classes of genes (signaling, microtubule-based movement, replication, DNA repair, and cell cycle) among the upregulated genes, while a different class of genes were downregulated during regeneration (cellular metabolism and processes related to translation, biogenesis, and the assembly of ribosomes). We observed similar trends in bisected *S. coeruleus*. More specific groups of genes found to be highly represented in the *S. polymorphus* data were the DYRK family of kinases and MORN domain proteins. We also find genes in these families to be represented among the upregulated genes during regeneration in *S. coeruleus*, both following bisection and following sucrose shock. In regenerating *S. polymorphus* anterior halves, the most highly expressed gene was reported to encode an ortholog of Lin-54, a DNA-binding regulator of cell cycle-related genes (*Schmit et al., 2009*). In our data, we identified a LIN54 ortholog in the sucrose shock response, but not in regenerating anterior half-cells. In regenerating posterior halves, *Onsbring et al., 2018*, reported a putative E2F/DP family member as highly upregulated, consistent with our findings. Comparing differentially expressed genes in *S.*

*coeruleus* (considering both sucrose and bisection) with their reciprocal best hits in the *S. polymorphous* proteome identified 365 genes upregulated in *S. polymorphous* that corresponded to genes upregulated in *S. coeruleus* (**Supplementary file 7**). The majority of these genes (325) were upregulated in either the posterior only, or the posterior and anterior, of *S. polymorphous*. When we specifically compared genes upregulated in the anterior or posterior halves of bisected *S. coeruleus* with genes specific to the anterior or posterior halves of bisected *S. polymorphous*, the correlation was not statistically significant (p=0.36 by Fisher's exact test). Thus, while similar families of genes are expressed in both species, they often seem to be expressed in different halves of bisected *S. coeruleus* and *S. polymorphous* cells.

*Wei et al., 2020*, analyzed OA regeneration using the chaotropic agent urea to induce OA shedding rather than sucrose shock as was done in our experiments. They found one cluster of genes that were downregulated, and two clusters of genes that were upregulated after urea shock. One of the upregulated clusters consisted largely of heat shock proteins and chaperones, which differs from our results showing that such genes tend to be downregulated during regeneration. A second upregulated cluster included EF hand proteins as well as GAS2, similar to our results, but unlike our findings with sucrose shock, they only found a single gene annotated as encoding a basal body or cilia-related protein, namely WDR90. The lack of cilia-related genes is consistent with what we observed in our OA-specific module, but differs from what we observed in our sucrose shock response data. Taken together, a comparison of our results with those of *Wei et al., 2020*, suggests that urea shock may induce a different regeneration program than sucrose, possibly involving a higher degree of cellular stress as indicated by the upregulation of stress response proteins in their experiments.

## Comparison with animal regeneration

Most studies of regeneration in animals focus on regeneration of tissues or limbs, which entails stem cell proliferation and differentiation to replace missing cell types. In order to ask whether there might be any similarities in upregulated genes between our studies of single-cell regeneration in *Stentor* and multi-cell regeneration in animals, we compared our results to gene expression patterns observed during regeneration in the planarian *Schmidtea mediterranea* (*Wenemoser et al., 2012*). Considering planarian genes for which a *Stentor* ortholog was identified, 19% of genes upregulated in planarian regeneration were also upregulated during *Stentor* regeneration (**Supplementary file 8**). To determine the significance of this fraction, we note that our study (**Supplementary files 1–6**) shows a total of 4775 genes with differential expression during regeneration in *Stentor* across the different modules, out of a total of 34,494 gene models in the *Stentor* genome. Thus, the probability that a randomly chosen *Stentor* gene is differentially expressed is approximately 14%. Thus the number of upregulated *Stentor* orthologs of planarian upregulated genes is comparable to that expected by chance, indicating no significant correlation between genes in the two studies. This result is not surprising given the nature of the upregulated genes in planaria, which consist almost entirely of genes related to cell proliferation, cell migration, or tissue organization, none of which occur during regeneration of single *Stentor* cells.

While most studies of regeneration in animals focus on regeneration of tissues by proliferation of stem cells whose progeny replaces the lost tissue, regeneration of single damaged cells can also take place in animals. One well-studied example is the case of nerve regeneration, in which a part of the cell (the axon) has to regrow after being severed. This form of regeneration is conceptually much more similar to the intracellular regeneration we observe in *Stentor*. We used the Regeneration Rosetta database (*Rau et al., 2019*) to compare expression of genes during *Stentor* regeneration to genes expressed during regeneration of the optic nerve in zebrafish (*Dhara et al., 2019*; see Materials and methods). Out of 606 zebrafish genes included in the study for which reciprocal best hits could be found in the *Stentor* genome, 110 (19%) were differentially expressed during regeneration in *Stentor* (**Supplementary file 9**). Again, this fraction is approximately what we would expect by chance, given the fraction of the *Stentor* genome (14%) that is differentially expressed during regeneration.

*Stentor* and optic neurons are both specialized cell types, whose regeneration program is dominated by the assembly of cell-specific structures such as the OA or neuronal growth cone. In particular, *Stentor* regeneration involves a large number of genes related to cilia and basal bodies, whose formation is not part of optic nerve regeneration. Genes of the Bisection and General classes do not include these OA-specific genes and might therefore show more overlap with animal cell regeneration. The

123 genes in these two categories constitute 0.3% of all *Stentor* genes, but account for 7 of the 606, or 1.1%, of *Stentor* orthologs of upregulated zebrafish genes. This greater than threefold difference in the fraction of upregulated orthologs relative to the fraction expected by chance is statistically significant (Z-test p=0.00024), consistent with a potential correlation, but the numbers of genes are very small.

One of the challenges with relating lessons learned in *Stentor* with other systems is that the annotation of the *Stentor* genome is still at an early stage. Our results will help with annotation by defining a subset of genes potentially involved in cellular morphogenesis.

## Cell cycle genes expressed during regeneration

One particularly notable class of genes expressed during regeneration are genes encoding cell cycle-related proteins. This has also been observed in previous reports (*Sood et al., 2017*; *Onsbring et al., 2018*; *Wei et al., 2020*). Examples include not only general cell cycle regulatory machinery such as cyclins and CDKs, but also mitotic kinetochore proteins. In order to obtain a more global view of cell cycle-related genes, we compared the genes upregulated during *Stentor* regeneration to a compilation of mouse genes showing cell-cycle-specific upregulation based on single-cell RNAseq analysis (*Kowalczyk et al., 2015*). We identified 107 *Stentor* genes having differential expression during regeneration that corresponded to mouse genes showing cell-cycle-specific gene expression (*Supplementary file 10*). Almost two-thirds of the *Stentor* upregulated genes, 61 in total, corresponded to genes expressed during S phase in mouse cells. Thirty-three upregulated *Stentor* genes corresponded to genes expressed during G2M, and the remaining 13 corresponded to G1-specific genes in mouse cells.

One hypothesis to explain expression of cell cycle genes during regeneration is that the cell cycle machinery might help regulate the timing of regeneration. The morphological steps of OA regeneration visible on the cell surface (*Figure 1*) are virtually identical to the steps by which a new OA forms during normal cell division (*Tartar, 1961*). Likewise, the macronucleus of *Stentor* undergoes a similar set of morphological changes during both division and regeneration. Like other ciliates, *Stentor* contains a single large polyploid macronucleus that contains approximately 50,000 copies of the expressed genome as well as several smaller diploid micronuclei. During division, the macronucleus is simply pinched in half by the cleavage furrow. Prior to this pinching, the elongated macronucleus shortens and compacts into a more spheroidal shape, which then re-elongates just before cytokinesis. These same shape changes occur during regeneration, even though the cell is not going to divide (*Paulin and Brooks, 1975*). The strong morphological similarity between regeneration and division, at both the cortical and nuclear level, together with the expression of cell cycle and mitosis-related genes during regeneration, suggests the possibility that OA regeneration might involve co-option of parts of the cell cycle machinery to regulate the timing of events. This potential connection between regeneration and cell division in *Stentor* highlights another classical question in the biology of regeneration: is regeneration a distinct process in its own right, or instead does it reflect a re-activation of development? In the case of the unicellular *Stentor*, for which development is equivalent to cell division, the use of cell cycle machinery in regeneration would support the latter view. The fact that knockdown of E2F leads to a delay or failure of late stages of regeneration may in part reflect a role of cell cycle-related E2F targets in this system. Indeed, a cyclin, cyclin A-associated protein, and Akt kinase are among the predicted E2F targets upregulated during regeneration in *Stentor*. We speculate that the delay phenotype in E2F RNAi experiments may result from reduced progression through cell cycle transitions associated with successive steps of regeneration.

However, an alternative explanation for the upregulation of cell cycle genes during regeneration could be the fact that in *Stentor*, the micronuclei undergo mitosis during regeneration (*Guttes and Guttes, 1959*). Expression of cell cycle and mitosis genes during regeneration might thus simply happen in order to allow for mitosis of the micronuclei, and have nothing to do with the events of regeneration itself. Consistent with this alternative hypothesis, we find that cell cycle genes are not among the genes upregulated during regeneration of tails in anterior half-cells, and it is known that neither micronuclear mitosis nor the macronuclear compaction associated with division occur during this process (*Weisz, 1949*; *Guttes and Guttes, 1959*).

The two explanations need not be mutually exclusive – it is possible that the program of OA regeneration is under control of cell cycle timing machinery that also directs the events of micronuclear

mitosis. In keeping with this idea, it has been shown that inhibition of the cell cycle-related Aurora kinases, several orthologs of which show differential expression during OA regeneration (clusters 2, 4, and 5 of the sucrose shock response) can advance or delay the later stages of regeneration (*Lin et al., 2020*).

## Implications for OA assembly

The OA is a complex cellular structure consisting of a tightly packed and highly ordered array of basal bodies and their associated cilia in the form of an MB together with associated structures involved in food ingestion. Out of 215 protein identified as highly abundance components of isolated MBs (*Lin et al., 2022*), 173 corresponded to genes upregulated during regeneration, while the other proteins were not detected in this study.

Given that OA formation requires synthesis of basal bodies, we would expect a priori that among the list of OA-specific regeneration genes will be genes encoding protein components of centrioles/ basal bodies, and that the expression of such genes should coincide with the time period in which the basal bodies are forming. Indeed, the OA-specific module does contain many known centriole biogenesis factors, and by far the majority of centriole-biogenesis genes are expressed in cluster 2 of the OA-specific pathway, which coincides with the period of basal body synthesis during OA regeneration. In the sucrose shock response pathway, we see the same trend of centriole-related genes expressed during cluster 2, and we also see cilia genes expressed in later clusters, coinciding with the timing of ciliogenesis and establishment of ciliary motility (*Figure 2E*). We thus find strong temporal correlation between the expression of centriole and cilia-related genes and the corresponding events of basal body biogenesis and ciliary assembly, respectively.

Based on this positive correlation with known genes involved in centrioles and cilia, we predict that at least some of the genes in these clusters with no or poor homology to known genes may encode undiscovered factors involved in centriole biogenesis and ciliogenesis. In particular, while the proteome of the centriole/basal body has become increasingly well characterized (*Keller et al., 2005*; *Jakobsen et al., 2011*; *Lauwaet et al., 2011*; *Firat-Karalar et al., 2014*; *Hamel et al., 2017*), we hypothesize that cluster 2 may contain genes whose products are needed for basal body assembly, or for positioning the OA in relation to other cellular structures, but may not encode structural components of the basal body itself. Such proteins would have been missed in proteomic analyses of the final structure. Interestingly, while many centriole/basal body-related genes are expressed in the early clusters (2 and 3), the only clear centriole-related genes we find in the late-expressed cluster 4 is an ortholog of LRRC45 which is a linker component required for centriole cohesion (*He et al., 2013*). In this regard, we note that when basal bodies first assemble during oral regeneration, they do so with random orientations relative to each other, creating a so-called 'anarchic field' (*Bernard and Bohatier, 1981*). It is only later in the process that the basal bodies associate into pairs and then larger groups to form the membranelles that are the dominant ultrastructural motif of the OA. The expression of LRRC45 at exactly this stage suggests that this linker may be a key element for assembling the MB from the initially randomly oriented basal bodies.

While we have noted that the OA-specific module has relatively few genes encoding ciliary proteins compared to the sucrose shock response dataset, it does contain a number of orthologs of genes implicated in the ciliopathies Meckel syndrome and Joubert syndrome. Both of these syndromes involve defects in non-motile cilia, and the proteins encoded by the Meckel and Joubert syndrome genes are involved in gating the import of proteins into the cilium (*Takao and Verhey, 2016*). The sucrose shock-specific program also included two Meckel/Joubert syndrome genes (B9D1 in cluster 2 and NPHP3 in cluster 4). We hypothesize that these genes may be expressed during regeneration in order to equip the newly formed basal bodies with appropriate protein machinery to generate OA-specific cilia.

In addition to basal bodies and cilia, the OA is also known to contain a set of protein fibers made of centrin-like EF hand proteins (*Huang, 1973*; *Lin et al., 2022*). The OA-specific module contains several predicted EF hand proteins including at least one ortholog of conventional centrin (see *Supplementary file 1*).

Finally, we note that the late expressing clusters contain orthologs of GAS2, a protein involved in coupling actin and microtubule cytoskeleton in other organisms. GAS2 has been found in the MB proteome (*Wei et al., 2020*; *Lin et al., 2022*). Our results further support involvement of GAS2 in OA

assembly, and suggest it functions at a late stage in the regeneration process. We hypothesize that GAS2 may play a role in aligning the membranelles with the longitudinal microtubule bundles that define the ciliary rows on the cell body.

## Re-use versus new synthesis of ciliary proteins

The OA is an assemblage of motile cilia. The cilia of the OA are substantially longer and more densely packed than the body wall cilia. Upregulation of cilia-related genes during ciliogenesis has been reported in other ciliates, including *Tetrahymena* (*Soares et al., 1991*) and *Paramecium* (*Kandl et al., 1995*), but it is also seen in green algae (*Schloss et al., 1984*; *Stolc et al., 2005*), lower plants (*Tomei and Wolniak, 2016*), and vertebrate multiciliated epithelial cells (*Ross et al., 2007*; *Hoh et al., 2012*). Given the fact that cilia represent the most visibly obvious structure within the OA, it was expected that regeneration of the OA would be accompanied by upregulation of genes encoding ciliary proteins. Consistent with this expectation, the sucrose shock-specific genes (*Figure 2C*) include a large number of genes encoding protein components of motile cilia, including radial spokes and the dynein regulatory complex. These genes are present mainly in clusters 3 and 4, and reach peak expression at 200 min (*Figure 2E*). It was therefore somewhat surprising to observe that genes encoding proteins specific to motile cilia, such as dynein arms or radial spoke proteins, are for the most part not found among the OA-specific genes (*Figure 2A*), indicating that these genes are expressed when the OA forms in sucrose-shocked cells, but not when the OA forms in posterior half-cells that are regenerating a new anterior half. Prior studies have found that full ciliary motility, including metachronal waves, is achieved in the OA at the same stage in development when the oral primordium has reached its full length, regardless of whether regeneration was induced by sucrose shock (*Paulin and Bussey, 1971*) or by surgical cutting (*Tartar, 1963*; *James, 1967*). Thus, the difference that we observe in terms of ciliary gene expression does not reflect a difference in the timing at which ciliary motility is recovered.

Given that the regenerating OA in posterior half-cells needs to be equipped with cilia, how is it possible that the genes encoding ciliary proteins are not, by in large, upregulated during regeneration? One possibility is that in these cells, assembly of motile cilia onto the basal bodies is carried out using protein obtained from either a pre-existing cytoplasmic pool or else from the pre-existing cilia on the cell body. *Schmähl, 1926*, has reported that in the giant ciliate *Bursaria*, some body cilia shorten while others are growing, suggesting an ability to redistribute protein between old and new structures. In the green alga *Chlamydomonas*, severed flagella are able to regenerate using protein from a cytoplasmic pool, but, importantly, they can also 'borrow' protein from other flagella present on the same cell, with those other flagella shortening as a result (*Coyne and Rosenbaum, 1970*). As to why OA assembly in sucrose shock cells involves expression of cilia-related genes while OA assembly in regenerating posterior half-cells does not, we speculate that scaling of organelle size may be involved. As *Morgan, 1901b*, has pointed out, when a cell is bisected and the posterior half forms an OA, the MB of the new OA is half the size of that in an intact cell. In contrast, a sucrose-shocked cell has to build a full-sized OA. It is therefore possible that the larger size of the OA being formed after sucrose shock requires synthesis of new protein, existing pools being insufficient. On the other hand, a bisected cell would start out with only half as much protein as an intact cell, so it is unclear if this type of scaling argument can really explain the differences that are seen in the transcriptional program. Clearly, direct assays for protein re-utilization will be needed to answer this question in the future.

## Implications for cellular wound recovery

The bisection-specific genes (*Figure 3B*) are shared between half-cells that are regenerating distinct structures. What the two half-cells have in common is that they were mechanically wounded during the surgical bisection, unlike the sucrose-shocked cells. We therefore interpret the bisection-specific differential gene expression pattern as reflecting a response to physical wounding of the cell. Direct measurements of membrane integrity in *Stentor* indicate that the plasma membrane seals itself on a time scale of 100–1000 s after wounding (*Zhang et al., 2021*). In comparison, the upregulated genes in the bisection response peak at 120–240 min after wounding, long after the wound itself has been closed. Thus, the bisection-specific transcriptional module is more likely to reflect recovery of cell state after wound closure, rather than the wound closure process itself.

Two of the upregulated genes in the bisection-specific module encode ammonium transporters, and a third encodes carbonic anhydrase. In animals, ammonium transporters and carbonic anhydrase

work together to control pH of both blood and cytoplasm (*Weiner and Verlander, 2019*). This is true not only in the kidney but also in cells such as muscle cells and macrophages (*Vaughan-Jones and Spitzer, 2002*; *Sedlyarov et al., 2018*). Changes in cytoplasmic pH have been reported following plasma membrane wounding or other forms of damage in animal cells (*Chambers and Chambers, 1961*). Since *Stentor* grows in pond water that is relatively acidic, a wound in the plasma membrane is expected to cause acidification of the cytoplasm. Carbonic anhydrase may act to increase the pH of cytoplasm after wound healing by dehydrating carbonic acid to $CO_2$. We infer that a key function of the bisection-specific transcriptional module is restoration of proper intracellular pH once the membrane rupture has been healed. Similarly, the expression of an ABC transporter and a major facilitator superfamily member, both of which are involved in transporting a wide range of small molecules out of cells (*Wong et al., 2014*; *Quistgaard et al., 2016*), may indicate a role in pumping contaminants out of the cell that may have entered through the open wound. Such pumping would be analogous to the use of a bilge pump to remove water from a boat after patching the hull.

## Role of conserved developmental regulators in single-cell regeneration

Giant, complex cells like *Stentor* face many of the same morphogenetic challenges as animal embryos in the need for establishing body axes, creating and maintaining patterns, and ensuring that anatomical features are present in the correct positions (*Marshall, 2020*). It is usually assumed that the similarity of such processes such as axiation, regeneration, or induction between unicellular protists and animal embryos must reflect analogous processes that are implemented using completely different, non-homologous mechanisms. The identification of Pumilio, a highly conserved developmental regulator (*Wreden et al., 1997*; *Gamberi et al., 2002*; *Sonoda and Wharton, 1999*) in the *Stentor* regeneration program, suggests that there may in fact be conserved molecular mechanisms at work during morphogenesis in both single-celled and multi-celled organisms.

Although multiple Pumilio orthologs were among the sucrose shock-specific gene list, this list did not contain any additional Pumilio targets besides those identified in the OA-specific list. Likewise, although the tail regeneration program contains Pumilio orthologs, it does not contain any predicted Pumilio target genes. One potential explanation is that some of the relevant Pumilio targets in these other programs may be genes whose transcripts are already present prior to initiation of regeneration. In such cases, expression of Pumilio orthologs may alter the localization or translation of those targets, even if they show no change at the transcriptional level.

## Conclusion

The transcriptional analysis *of Stentor* regeneration described here begins to reveal key molecular details of intracellular patterning and regeneration mechanisms, such as evidence for modularity and a cascade organization. We find that *Stentor* regeneration involves expression of regulatory genes conserved across eukaryotes, suggesting a deep conservation of developmental mechanisms.

## Materials and methods
### Stentor culture and regeneration

Cells were obtained from Carolina Biological Supply (cat. num. 131598) and cultured as previously described (*Slabodnick et al., 2013*). Cells were maintained in Pasteurized Spring Water (Carolina Biological Supply) and fed with *Chlamydomonas* and wheat seeds. Cells were collected from the same culture for each RNAseq experimental replicate.

To induce regeneration of the OA by sucrose shock (*Lin et al., 2018*), cells were placed in a 15% sucrose solution for 2 min (*Tartar, 1957*), and then washed in Carolina Spring Water thoroughly. Samples of ~20 cells were collected before shock, then at 30 min post shock, 1, 2, 3, 4, 5, 6, 7, and 8 hr. At each time point, a sample of cells was lysed into RNA-stabilizing buffers specified by the extraction kit, and then stored on ice until the end of the experiment when the RNA purification was performed in parallel on all samples (see below). Four replicates were analyzed for each time point.

For analysis of regeneration in bisected cells, cells were cut in half as previously described (*Lin et al., 2018*). A 50 µl drop of methylcellulose was placed onto a slide, and 40–50 *Stentor* cells were collected in a volume of 50 µl and added to the drop of methylcellulose. Cells were individually cut with a glass needle (*Lin et al., 2018*), making sure to complete all cutting within 10 min from the time

the first one was started. As a result, all samples are synchronous to within 10 min of each other. The anterior and posterior half-cells were manually separated into two tubes and washed once with spring water to remove the methylcellulose. Samples were then incubated at room temperature for the specified time period (60, 90, 120, 180, 360, or 420 min). Samples designated as t=0 were collected within 2 min prior to cutting. After the desired time had elapsed, the media was removed from the cells to produce a final volume of less than 20 µl, and 350 µl of RLT buffer from Qiagen micro easy kit was added to the sample and mixed by pipetting 20 times. Lysate for each sample was stored at 4°C while additional samples were prepared.

## Total RNA extraction

RNA was extracted at each time point using the Nucleospin RNA XS kit from Clontech (cat. num. 740902.250). RNA quality was assessed using a NanoDrop and then Bioanalyzer was used to quantify RNA amount. ERCC spike ins (Thermo Fisher, cat. num. 4456739) were added to each sample in a dilution ranging from 1:1000 to 1:10,000 depending on the initial amount of RNA extracted.

## RNAseq library preparation and sequencing

RNAseq libraries were prepared with Ovation RNAseq system v2 kit (NuGEN). In this method, the total RNA (50 ng or less) is reverse transcribed to synthesize the first-strand cDNA using a combination of random hexamers and a poly-T chimeric primer. The RNA template is then partially degraded by heating and the second-strand cDNA is synthesized using DNA polymerase. The double-stranded DNA is then amplified using single primer isothermal amplification (SPIA). SPIA is a linear cDNA amplification process in which RNase H degrades RNA in DNA/RNA heteroduplex at the 5'-end of the double-stranded DNA, after which the SPIA primer binds to the cDNA and the polymerase starts replication at the 3'-end of the primer by displacement of the existing forward strand. Random hexamers are then used to amplify the second-strand cDNA linearly. Finally, libraries from the SPIA amplified cDNA were made using the Ultralow v2 library kit (NuGEN). The RNAseq libraries were analyzed by Bioanalyzer and quantified by qPCR (KAPA). High-throughput sequencing was done using a HiSeq 2500 instrument (Illumina). Libraries were paired-end sequenced with 100 base reads.

RNAseq data from this study are available in GEO under accession number GSE186036.

## RNAseq data preparation – trimmed and filtered reads

We used trimmomatic (*Bolger et al., 2014*) to trim RNAseq reads with the following flags: ILLUMI-NACLIP:$adapterfile:2:30:10 HEADCROP:6 MINLEN:22 AVGQUAL:20 The settings ensured that we kept reads of at least 22 bases, an average quality score of 20 and trimmed any remaining Illumina adapter sequences.

## Transcriptome generation

To generate a transcriptome, we combined all the reads from all RNAseq samples and time points. We ran Tophat2 (*Kim et al., 2013*) to align the reads to the genome (StentorDB http://stentor.ciliate.org; *Slabodnick et al., 2017*). We used the following flags to ensure proper mapping in spite of Stentor's unusually small introns (*Slabodnick et al., 2017*): -i 9 -I 101 `--min-segment-intron 9 --min-coverage-intron 9 --max-segment-intron` 101 `--max-coverage-intron` 101p 20.

We then ran Trinity (*Haas et al., 2013*) using a genome guided approach. We used the following flags: `--genome_guided_max_intron` 1000.

In total, we assembled 34,501 genes aligned with the existing *Stentor* genome models in StentorDB, plus 143 novel models. Our prior gene prediction for the *Stentor* genome indicated approximately 35,000 genes were present (*Slabodnick et al., 2017*). Such a large number of genes is typical of ciliates.

## Calculating transcript abundance and differential expression analysis

We used Kallisto to quantify transcript abundance (*Bray et al., 2016*) using the following flags: -t 15 -b 30. We then used Sleuth (*Pimentel et al., 2017*) to identify genes which are differentially expressed genes through the regeneration time course. We use an approach similar to that used by Ballgown (*Frazee et al., 2015*). We fit the expression data to time using natural splines (R function 'ns') with 3 degrees of freedom. Then, using Sleuth, we compared this model to a null model where change

in expression is only dependent upon noise. To decide if transcripts were differentially expressed, we defined the minimum significance value (qval in the Sleuth model) to be 10 times the minimum significance value of all the ERCC spike-in transcripts. We found that nearly 5583 transcripts are differentially expressed during OA regeneration. Of these, 485 had no clear homology to proteins in NCBI databases nor PFAM. We identified 234 that did not map to existing gene models. We averaged the expression of all transcripts that mapped to gene models as well as those which were part of a Trinity transcript cluster. All subsequent analysis was performed on these averaged values. Clustering analysis was performed as follows. First, genes whose maximum expression among the post-shock time points was found 30 min after sucrose shock were put into one cluster manually. Gene expression profiles before sucrose shock and 30 min after are highly correlated (correlation coefficient from Pearson's correlation = 0.99). Then, the remaining genes were clustered into four groups using 'clara'.

Heatmaps were generated using all transcripts identified, including those that do not have matches in the current set of gene models in StentorDB. The gene lists given in the supplementary files, as well as our discussion of results, are restricted to those transcripts for which corresponding gene models exist in the genome database. For this reason, the number of rows in each heatmap will in general be slightly larger than the number of genes listed in the corresponding tables.

## Annotation of transcriptome

Following the approach of trinotate (https://trinotate.github.io), we annotated the transcriptome. First we used transdecoder (http://transdecoder.github.io/) to find the longest ORFs (minimum protein length is 100AA and uses the standard genetic code). We used blastx and blastp (*Altschul et al., 1990*) to search the Uniprot database (*Uniprot consortium, 2017*). Then Hmmscan (hmmer. org, HMMER 3.1b1) was used to search the pfam-a database (*Finn et al., 2016*). Alignments of genes of interest were further manually inspected using a blastp search against the 'Model Organism' or 'Uniprot-KB/Swiss-Prot' databases.

## Mapping transcripts to gene models and to genome

We used Gmap (*Wu and Watanabe, 2005*) to map transcripts to gene models following the approach outlined here: https://github.com/trinityrnaseq/RagonInst_Sept2017_Workshop/wiki/genome_guided_trinity.

We used a built-in script from Trinity to utilize Gmap to align transcripts to a repeat-masked (rmblastn 2.2.27+) Stentor genome. We used bedtools (*Quinlan, 2014*) on the resulting bam file to identify overlaps between the aligned transcripts and existing gene models (*Slabodnick et al., 2017*). Annotations from StentorDB were collected, most of which refer to other ciliate genomes. For any gene with a predicted domain or a ciliate homolog, BLAST search was performed against the *Chlamydomonas* genome version 5.5 on the JGI Phytozome database.

## Annotation of subsets of genes

Because centriole and cilia genes have often been poorly annotated in existing databases, we manually curated 'ancestral centriole genes' (*Azimzadeh et al., 2012*) and other genes involved with ciliogenesis and centriole biogenesis (*Ishikawa and Marshall, 2011*; *Bettencourt-Dias and Glover, 2007*). We used a reciprocal best BLAST search approach to identify genes in the *Stentor* genome with homologs to these manually curated sets of genes. Other gene categories (nuclear, RNA binding, etc.) were annotated based on existing annotations in the Stentor genome database.

In order to identify potential targets of E2F and Pumilio, we searched the non-coding sequence of the *Stentor* genome using Perl with the following regular expressions:

E2F: TTT[GC][GC]CGC
Pumilio: TGTA[CTAG]ATA

## Bioinformatic analysis of gene sets in other organisms

For comparison with *S. polymorphus*, reads for *S. polymorphus* from *Onsbring et al., 2018*, were used to assemble a transcriptome using FASTQC, trimmomatic, CutAdapt, and Bowtie2, followed by Kallisto and DESeq2 to obtain read counts. Genes showing significant differential expression were

clustered and the reciprocal best hit *S. coeruleus* genes identified. These results are tabulated in *Supplementary file 7*.

For comparison with expression patterns in animal regeneration, we used the PlanMine database to find the *S. mediterranea* gens in the Smed version 3 genome corresponding to genes listed in Table 1 of *Wenemoser et al., 2012*, and then searched each for a corresponding *Stentor* ortholog using blastx search in StentorDB. For comparison with expression during optic nerve regeneration in zebrafish, we used the Regeneration Rosetta database to select genes showing differential expression at the 5% confidence level. We then used reciprocal best hits with human genes to identify the *Stentor* orthologs of upregulated zebrafish genes, which indicated 606 zebrafish genes differentially expressed during optic nerve regeneration for which a *Stentor* ortholog could be identified. Matching the *Stentor* gene IDs of these 606 genes indicated that 110 of them corresponded to *Stentor* genes differentially expressed during regeneration.

## RNAi analysis of E2F and Pumilio

RNAi by feeding was performed as previously described (*Slabodnick et al., 2014*). Briefly, RNAi experiments are set up in 12-well plates with 2 ml of filtered spring water in each well. Feeding is performed under a Zeiss Stemi dissecting microscope. Bacteria expressing the RNAi constructs were added to the well daily for 10 days, exchanging the media for fresh media half way through the experiment. Images were acquired using a Zeiss Axiozoom microscope with a ×2.3 objective lens and total magnification of ×40 or ×80 using a Nikon Rebel T3i digital SLR camera. The presence, position, and size of the OA are assessed by visualizing the MB, which is detected in living cells based on the visualization of a dense array of long, beating cilia. This is in contrast to the body wall cilia which are much shorter and not visible at the magnification used in our imaging. The OA primordium is detected as a flat region on the otherwise curved side of a cell, often with a decreased opacity of the adjacent cytoplasm. High magnification video imaging has shown that this flat patch with a flanking transparent zone is a reliable indicator of the primordium.

For the E2F experiments summarized in *Figure 5K*, regeneration on schedule was defined by the presence of a visible OA primordium on the side of the cell by 4 hr, an OA that contains an elongated MB with visible beating cilia by 6 hr, an OA with motile cilia that has migrated to the anterior end of the cell by 8 hr, and cells that are still alive and have an OA with motile cilia encircling the anterior end.

Constructs were made using plasmid pPR-T4p and transformed into *Escherichia coli* cell line HT115 for expression of dsRNA. Inserts were cloned by PCR with the following primers, with the overhangs needed for T4 ligation-independent cloning indicated in underline.

E2F (SteCoe_12750):

5' <u>CATTACCATCCCG</u>CATAAAAGCCACGGCTCATC 3'
5' <u>CCAATTCTACCCG</u>GAGCAAAGATCAGGGTCAGG 3'

Pumilio (SteCoe_27339):

5' <u>CATTACCATCCCG</u>AAGGTGATGATGCTGATG 3'
5' <u>CCAATTCTACCCG</u>CCAAGCAATTCAAAACATGC 3'

## Acknowledgements

The authors thank Jennifer Morgan, Jeremy Reiter, Robert Stroud, Hiten Madhani, and members of the Marshall lab for many helpful discussions, and Vincent Boudreau, Sindy Tang, and Mark Slabodnick for helpful comments on the manuscript. We thank Jeremy Reiter for providing a table of human reciprocal best hits for Stentor. This work was supported by an American Cancer Society postdoctoral fellowship (PS), NSF predoctoral fellowships (AL and CY), an HHMI Gilliam Fellowship (UD), NSF grant MCB-1938109 (SKYT), and NIH grant R35 GM130327 (WFM). Initial stages of this work were supported by the UCSF Program in Breakthrough Biomedical Research (WFM).

## Additional information

### Funding

| Funder | Grant reference number | Author |
|---|---|---|
| National Institutes of Health | R35 GM130327 | Wallace F Marshall |
| National Science Foundation | MCB-1938109 | Sindy KY Tang Wallace F Marshall |
| American Cancer Society postdoctoral fellowship | | Pranidhi Sood |
| NSF predoctoral fellowships | | Athena Lin Connie Yan |
| HHMI Gilliam Fellowship | | Ulises Diaz |

The funders had no role in study design, data collection and interpretation, or the decision to submit the work for publication.

### Author contributions

Pranidhi Sood, Conceptualization, Data curation, Formal analysis, Investigation, Methodology, Writing - original draft; Athena Lin, Validation, Investigation, Methodology; Connie Yan, Tatyana Makushok, Ambika V Nadkarni, Investigation; Rebecca McGillivary, Investigation, Visualization; Ulises Diaz, Data curation, Formal analysis, Investigation; Sindy KY Tang, Supervision, Funding acquisition; Wallace F Marshall, Conceptualization, Data curation, Software, Supervision, Funding acquisition, Visualization, Writing - original draft, Project administration, Writing - review and editing

### Author ORCIDs

Connie Yan ⓘ http://orcid.org/0000-0002-9961-0671
Wallace F Marshall ⓘ http://orcid.org/0000-0002-8467-5763

### Decision letter and Author response

Decision letter https://doi.org/10.7554/eLife.80778.sa1
Author response https://doi.org/10.7554/eLife.80778.sa2

## Additional files

### Supplementary files

• Supplementary file 1. Genes showing differential expression during oral apparatus regeneration in both sucrose-shocked and bisected cells.

• Supplementary file 2. Genes showing differential expression during oral apparatus regeneration in sucrose-shocked, but not bisected, cells.

• Supplementary file 3. Genes showing differential expression during regeneration in posterior half-cells.

• Supplementary file 4. Genes showing differential expression during regeneration in anterior half-cells.

• Supplementary file 5. Genes showing differential expression during regeneration in both posterior and anterior half-cells, but not in sucrose shock.

• Supplementary file 6. Genes showing differential expression during regeneration in all three samples analyzed: sucrose shock, anterior half-cells, and posterior half-cells.

• Supplementary file 7. Comparison of differential gene expression during regeneration in *Stentor coeruleus* and *Stentor polymorphous*.

• Supplementary file 8. Comparison of differential gene expression during regeneration in *Stentor coeruleus* and planarian flatworms.

• Supplementary file 9. Comparison of differential gene expression during regeneration in *Stentor coeruleus* and zebrafish optic nerve.

• Supplementary file 10. Comparison of differential gene expression during regeneration in *Stentor*

*coeruleus* and differential gene expression in the mammalian cell cycle.

• MDAR checklist

## Data availability

transcriptomic data have been deposited in GEO under accession code GSE186036.

The following dataset was generated:

| Author(s) | Year | Dataset title | Dataset URL | Database and Identifier |
|---|---|---|---|---|
| Sood P, Marshall W, Lin A, McGillivary R | 2021 | Modular, Cascade-like Transcriptional Program of Regeneration in Stentor | https://www.ncbi.nlm.nih.gov/geo/query/acc.cgi?acc=GSE186036 | NCBI Gene Expression Omnibus, GSE186036 |

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
