## [Editor Report]

This ground-breaking study builds on recent genome annotation to report the gene expression pattern that drives morphogenesis of Stentor, a large and beautifully organized single-celled organism with a remarkable ability to regenerate after damage. The study has been greatly strengthened by the addition of two molecular perturbation experiments that provide important insights into the regeneration process.

---

## [Decision Letter]

**Decision letter after peer review:**

[Editors’ note: the authors submitted for reconsideration following the decision after peer review. What follows is the decision letter after the first round of review.]

Thank you for submitting the paper "Modular, Cascade-like Transcriptional Program of Regeneration in *Stentor*" for consideration by *eLife*. We profoundly apologize for the very long delay with the peer review of your article, which was due to the fact that the Reviewing Editor became unavailable. Your article has been reviewed by 3 peer reviewers, and the evaluation has been overseen by a Reviewing Editor and a Senior Editor. The following individuals involved in review of your submission have agreed to reveal their identity: Timothy J. Mitchison (Reviewer #1).

Comments to the Authors:

We are sorry to say that, after consultation with the reviewers, we have decided that this work will not be considered further for publication by *eLife*. While all three reviewers were very enthusiastic about the topic and the performed analyses, they all found that the paper was too preliminary for *eLife* due to the lack of functional data and spatial information. Obtaining such data would require extensive efforts that go beyond an *eLife* revision, and therefore, it seems that it would be better to submit this paper elsewhere.

*Reviewer #1 (Recommendations for the authors):*

Stentor is a marvel of single cell organization and regeneration. It is also a exemplar of complex single cell body plans that characterize protists which are completely different from more typical plant and animal cells. Understanding how Stentor is organized and how it regenerates is a grand challenge in biology. This paper is a competent step in that direction which builds on recent genome sequencing. The experimental design is elegant. The discussion is very thoughtful and interesting – the comparison of regeneration and division was particularly eye-opening for this reviewer.

This reviewer lacks the informatics experience to critically evaluate the data analysis, but assuming it is sound, we learn a lot about how Stentor regenerates.

There is satisfying extension of mechanistic hypothesis from previous morphological studies to the molecular level in some cases. In particular, this paper confirms that regeneration occurs in a stepwise fashion and that late steps require translation of genes induced at early steps. It extends this important idea by reporting immediate early vs later genes and mapping them onto structures that are built at different stages. There is satisfying categorization of early and late induced genes into classes that map to early and late regenerating structures. These observations, combined with the arrest of the morphogenetic and transcriptional program by cycloheximide, constitutes the main mechanistic discovery in the study.

The paper makes two sets of interesting cause-effect predictions at the molecular level, concerning E2F and puillio homologs, but tests neither. The cycloheximide experiment is a molecular perturbation. Its use to identify immediate early genes and distinguish them from translation dependent gene immediate is the strongest molecular discovery in the paper.

The one big aspect of the biology that is missing for this reviewer is any sense of spatial organization. The paper is all about temporal organization. Temporal cascades of gene expression downstream of a perturbation are ubiquitous in Eukaryotes. In that respect, the results of this paper are very familiar. Stentor is extraordinary in its spatial organization, but this aspect of the biology is not addressed. The paper left me desperately wanting to know where the induced genes are transcribed and translated. In particular, to what extent does overall spatial organization arise from self-organization (or directed transport) of proteins post-translation vs local programming of structure by local translation. Addressing that question might require in situ methods that presumably do not yet exist. It is not mentioned in the discussion whether this sort of question been addressed for any ciliate. Some recent methods for addressing local translation do not require in situ hybridization. For example, there are methods for imaging nascent proteins by measuring their proximity to the peptide bond forming site in ribosomes that might be applicable to Stentor.

One way the authors could begin to address spatial regulation without the need for new methods would be to trigger regeneration of the oral apparatus (possibly with and without cycloheximide), then bisect the animal at different time points and collect mRNA from proximal vs distal halves. They could then ask whether induced mRNAs are preferentially transcribed/translated in the half of the organism where the oral apparatus is assembling. It would be important to normalize carefully for total mRNA and ideally to distinguish spatially localized transcription vs mRNA localization. That might be possible by looking at introns. Such studies could be executed in an unbiased way using RNAseq or the authors could look at specific induced genes by qPCR.

The E2F hypothesis is very interesting. The paper would be obviously stronger if this provocative mechanistic hypothesis were tested. The same is true for the pumillio hypothesis. That said, to this reviewer, the issue of spatial regulation is more interesting. Starting to address spatial regulation would add more to the paper than addressing a molecular cause-effect prediction.

In the intro I feel the authors are trying too hard to frame Stentor as a model for cell wounding in general and fail to sufficiently highlight the amazing sub-cellular organization and regeneration capacity of ciliates in general and Stentor in particular. They also seem to blur cellular regeneration and tissue regeneration. This writing style presumably reflects NIH priorities. In this reviewer's opinion, ciliates are not a good model for human anything but will reveal completely new ideas on sub-cellular organization that underlie protist biology.

"…..by reducing bicarbonate to CO2" is incorrect chemistry. Should read "by dehydrating carbonic acid to CO2". No redox chemistry is involved.

The discussion should touch on the state of genome annotation in Stentor and specify the extent to which this study has improved it. Improving genome annotation is an important contribution and would increase the significance of the paper.

The comparison of regeneration and cell cycle progression in the discussion was really interesting to this reviewer. This set of points might be improved by adding a figure panel that identified clusters of genes that are common to both processes.

As a side note, it might be interesting to stain Stentor at different stages of regeneration and cell cycle progression with an excellent antibody to Phospho-TP (Cell Signaling Technology #9391S). This antibody brightly stains multiple structures in prophase and metaphase human cells and might reveal interesting aspects of spatial regulation by CDKs in Stentor

*Reviewer #2 (Recommendations for the authors):*

Stentor coeruleus is a large single-cell ciliate with high regenerative potential, and thus provides the opportunity for researchers to understand how single cells build and regenerate parts. In addition, comparisons with metazoan regenerative models will uncover both deeply conserved principles of regeneration and aspects unique to single cell regeneration. In this study, the authors use RNA-seq to comprehensively profile the transcriptional changes that occur over the course of three regeneration paradigms: (1) Oral apparatus (OA) regeneration that occurs after sucrose shock, (2) Regeneration of the "anterior end" (which includes the OA) after bisection, and (3) Regeneration of the "posterior end" after bisection. By comparing the transcriptional changes that occur during these three distinct regeneration paradigms, the authors were able identify modules that were specific to certain events, such as regeneration of the OA and the reaction to bisection. The authors use these data to identify some putative key regulators. For example, the authors identified E2F as a transcriptional regulator that could be required for OA regeneration, as well as several homologs of the conserved RNA regulator Pumilio with unique expression patterns during regeneration. Notably though, no candidate regeneration regulators were tested using the RNAi methods developed by this laboratory. Finally, RNA-seq was also performed for OA regeneration in the presence of a translation blocker, which allowed the authors to separate transcriptional responses that were initiated by existing proteins (largely the earlier transcriptional responses) from those that were dependent on the production of new proteins. Based on these data, the authors conclude that regeneration occurs as a cascade of transcriptional events, with one wave triggering the next. Although the authors discuss and an alternate regulatory architecture that could conceivably control regeneration in Stentor (the initial injury response individually activating each regeneration module directly), the "cascade" model seems expected in this case. Overall, the study provides a useful and comprehensive starting point to unraveling the molecular mechanism underlying Stentor regeneration. However, three shortcomings of this study are: (1) lack of functional data for identified candidates testing their requirement for regeneration, (2) the finding that regeneration occurs through a "cascade" of transcriptional events seems expected (if it is not expected, the authors would need to cite clarifying literature) and (3) the authors conclude that their data show "deep conservation of developmental mechanisms," which seems strong based on an expression study in which only a small number of possibly conserved genes were discussed in the text.

From the perspective of a developmental biologist, it is not surprising that regeneration occurs as a cascade of events (i.e. a GRN) triggered by an immediate early response that is translation-independent. It also doesn't seem surprising that regeneration of different parts would require different gene modules. However, this may be my own limited view. If there are examples from the literature that support the possibility of other regulatory architectures being used in, for example, cellular repair, these should be cited to help support the significance of the findings here.

It is certainly surprising/confusing that the OA regeneration specific module does not include cilia motility genes. Could an explanation be that it takes longer to regenerate cilia in the bisected animals and the time course didn't go long enough to capture that step?

Figure 1A,B – It would be helpful to label the Stentor parts and the A/P axis on the figure. Also, is the distribution of micronuclei known? After the bisection, does each half get at least one? Do more need to be made? Can a fragment of stentor regenerate without a micronucleus? These seem like important points in trying to understand how regeneration works. I'm not suggesting experiments should be done to address this point, but if the answers to some of these questions are known, I think it would be helpful for the reader to know.

Figure 1C – It would be helpful to define the colors in the figure legend. Also, be consistent with capitalization

Figure 2B – What method was used to place genes into the various categories? This information should be included in the methods section.

Figure 5: Are the putative E2F targets not upregulated in the regenerating tails? If so, this would further support the hypothesis that the list shown in Figure 5B are targets of E2F.

The methods should include a description of how putative E2F and pumilio binding sites were identified.

*Reviewer #3 (Recommendations for the authors):*

The manuscript by Sood et al. aims to describe transcriptional changes underlying regeneration in the large, single-celled ciliate Stentor, which is a fascinating and novel model for wound repair, the restoration of polarity, and regeneration. Stentor regeneration likely occurs through mechanisms distinct from those operating in metazoans which rely heavily on differentiation of new cell types. This is a new and exciting model organism for understanding regeneration and the transcriptional programs controlling this process. By comparing different regeneration paradigms (anterior and posterior regeneration, vs sucrose shock which removes only the oral apparatus), the authors aim to obtain a global view of the transcriptional changes that drive regeneration in Stentor. While the biology is potentially interesting, the paper over-relies on transcriptional profiling without any functional testing of the genes, pathways, or processes uncovered.

1. This manuscript lacks a connection between the transcriptional changes observed during regeneration and any biological outcomes. Without confirmation of these descriptive findings, it is difficult to understand the significance of these newly discovered transcriptional programs in driving regeneration. There are several potential ways that this could be at least attempted: RNAi, previously demonstrated to work in Stentor by the Marshall lab; pharmacological perturbations of key genes or pathways; or (at a bare minimum) clearer grouping of categories according to timing of gene expression or predicted function. The major genes noted in the abstract (E2F and Pumilio) are indeed important in other systems for basic cellular functions, but changes in their expression only hint at their potential function in regeneration.

2. Overall, the major conclusion is that regeneration occurs in distinct, transcriptional waves. This phenomenon has been observed in other regenerating systems (e.g. Wenemoser and Reddien, Genes & Dev 2012, which also uses cycloheximide treatment to identify translation-independent transcriptional changes). The analysis and the paper in its current form is overly focused on Stentor, and lacks placement in the context of regeneration. A hugely interesting aspect of studying regeneration in Stentor is that it occurs in single cells – and therefore is unlikely to require the cell division and fate specification that is known to drive regeneration in planarians, axolotls, and mammals. Comparing these differences would be worthwhile and would highlight any similarities that the authors might have observed.

3. The authors performed 3 different regeneration challenges, presumably to garner the transcriptional changes that are shared between all conditions, and those that are unique to each amputation. Because individual genes in the heat maps are not labeled, it's difficult to glean from the figures which genes fall into the general or specific categories, or are carried through from one figure to the next. Additionally, most of the heat maps are not labeled and therefore do not provide a clear picture of the transcriptional changes that they describe.

4. In Figure 4C, the text says that 21 genes are unchanged after cycloheximide treatment, but the figure panel has 30 rows. It is unclear whether each row correlates to a single gene. The data presented in this figure is not visually/conceptually connected to the identification of a set of immediate early genes as described in the text. If there is indeed a set of "immediate early" genes why are they not mentioned in the text? Furthermore, the statement that their products are required for subsequent transcriptional programs is not demonstrated by the data in the figure.

5. In Figure 5, the logic for selecting E2F is insufficiently conveyed. It is unclear whether E2F one of the more strongly upregulated genes in the cycloheximide-treated animals, or whether the authors are pursuing this as a candidate approach? What are the predicted targets of E2F, how were they identified, and where is the quality control verifying that these targets exist in Stentor? If possible, it would be good to name them in Figure 5B. If they are indeed targets of E2F, then a decrease in their transcription should be observed after cycloheximide treatment.

6. In Figure 6, there needs to be some analysis to support the significance of genes with pumilio binding sites. The conclusion that pumilio may be regulating signaling pathways because several of these genes are kinases/phosphatases is unconvincing.

7. In Figure 2, color coding of clusters should be revised to correlate logically with the message of the figure in 2B/2D (e.g. shades of a single color that get darker or lighter in successive clusters) to show a visual pattern, rather than orange and red representing the earliest/latest clusters.

8. In Figure 3A/3B, is there a difference between "bisected top" and "top bisection"

9. In general, the figures do not convey a clear message about specific components of the centrioles, although the results describe them, resulting in confusion about whether this is a major focus of the research or not. For example, Figure 2E would benefit, for the general reader, from a figure diagramming the biological significance of these two graphs, or even titles on top of the graphs to clarify what these different groups of genes represent. Similar for 2H/2I.

[Editors’ note: further revisions were suggested prior to acceptance, as described below.]

Thank you for resubmitting your work entitled "Modular, Cascade-like Transcriptional Program of Regeneration in Stentor" for further consideration by *eLife*. Your revised article has been evaluated by Anna Akhmanova (Senior Editor) and a Reviewing Editor.

The manuscript has been strongly improved and all reviewers are in principle supportive of publication, but there are some remaining issues, mostly concerning the writing and figure clarity, that need to be addressed, as outlined below:

*Reviewer #1 (Recommendations for the authors):*

The revised manuscript now includes knockdown phenotypes for E2F and Pumilio, which was one of the key deficiencies in the previously submitted version. Despite the addition of these phenotypes, the biological significance of the findings is not connected to the analysis of regeneration in a meaningful way to make a clear conclusion. Overall, many of the conclusions (in particular, whether regeneration relies on a cascade or parallel induction) rely on correlations rather than direct testing of these possibilities.

In Figure 4, the impact of cycloheximide on gene expression in all clusters is clear from the heat maps in panel D. The text states that the "largest effect" is in clusters 4 and 5. Can these changes be quantified in some way to make this assessment less subjective? This is important because cluster 1, which is upregulated at the earliest times during regeneration, is also affected, yet the authors make the conclusion that clusters 4 and 5 are most affected by blocking translation. Making this conclusion stronger is essential to attempt to distinguish between the two models in A. Also, what is the impact of cycloheximide treatment on regeneration?

Some conclusions in the Results section are difficult to extract from the figures. For example, is there data to back up this statement from the top of p. 15, regarding E2F: "In cells treated with cycloheximide, 12/13 of these targets fail to be expressed." If it is in Figure 5B, it should be clearly marked where the failure is, and at which time, or the gene expression should be assayed in an independent manner.

In E2F RNAi experiments (Figure 5), the panels should be labeled with time points and RNAi conditions. The methods should include a description of how these phenotypes were scored, because many readers of this paper may not be familiar with how to convincingly identify an OA.

The discussion now contains many more broad comparisons of gene expression trends in different contexts and the message is lost. It now reads more like a Results section and would benefit from a strong edit.

Figure 6F-H. The authors should include control images in addition to the RNAi animals. The y axis on H should be labeled. MB should be redefined since it's only mentioned once in the paper.

For all figures where regeneration is measured at different times, the y axis is labeled "regenerating on schedule". I appreciate that some quantification is always arbitrary but there should be some information in the methods about how this subjective measurement was made.

At the top of page 11, describing Figure 2G: "Cluster 1 represents down-regulated genes." is confusing, because these genes are initially upregulated but gradually decrease over 7 hours. Please rephrase.

Can the defect in OA regeneration observed in E2F RNAi that emerges between 4-6 hours after amputation, and potential E2F targets or genes that are expressed be connected?

*Reviewer #2 (Recommendations for the authors):*

One of my previous concerns was the lack of functional data, which has now been added. While these data have the potential to boost the significance of the study, there are issues with the data presentation outlined below:

E2F RNAi experiments: The images shown in Figure 5 panels C-J are not sufficiently labeled. It is unclear which panels are RNAi treated and which are controls. It is unclear which time points are shown. It is unclear what the arrow and arrowheads are pointing to. The images are not of high enough quality to make out any detailed features. The text asks the reader to look for the "OA primordium" in the pictures, but there are no discernible structures observed, at least to the untrained eye.

Pumilio RNAi experiments: The images shown of regenerating cells after pumilio RNAi seem to indicate a disruption or delay in regeneration, but there are issues with data presentation. First, no images of control regenerating cells are shown for comparison. The images are not of high enough quality for a naïve reader to identify the membranellar band (i.e. the structure that is being pointed to in Figure S1F by the purple arrow). In Figure 6H, an orange bar shows up with no explanation in the "spherical cell" category and the abbreviation "MB" should be explained in the legend (I had to go back up to the introduction to remind myself what that was).

Beyond the new RNAi data presented, the authors have responded to my other previous concerns satisfactorily.

---

## [Author Response]

[Editors’ note: the authors resubmitted a revised version of the paper for consideration. What follows is the authors’ response to the first round of review.]

Reviewer #1 (Recommendations for the authors):Stentor is a marvel of single cell organization and regeneration. It is also a exemplar of complex single cell body plans that characterize protists which are completely different from more typical plant and animal cells. Understanding how Stentor is organized and how it regenerates is a grand challenge in biology. This paper is a competent step in that direction which builds on recent genome sequencing. The experimental design is elegant. The discussion is very thoughtful and interesting – the comparison of regeneration and division was particularly eye-opening for this reviewer.This reviewer lacks the informatics experience to critically evaluate the data analysis, but assuming it is sound, we learn a lot about how Stentor regenerates.There is satisfying extension of mechanistic hypothesis from previous morphological studies to the molecular level in some cases. In particular, this paper confirms that regeneration occurs in a stepwise fashion and that late steps require translation of genes induced at early steps. It extends this important idea by reporting immediate early vs later genes and mapping them onto structures that are built at different stages. There is satisfying categorization of early and late induced genes into classes that map to early and late regenerating structures. These observations, combined with the arrest of the morphogenetic and transcriptional program by cycloheximide, constitutes the main mechanistic discovery in the study.

We greatly appreciate the supportive comments from the reviewer, and have tried our best to improve the manuscript based on the constructive suggestions.

The paper makes two sets of interesting cause-effect predictions at the molecular level, concerning E2F and puillio homologs, but tests neither. The cycloheximide experiment is a molecular perturbation. Its use to identify immediate early genes and distinguish them from translation dependent gene immediate is the strongest molecular discovery in the paper.

We have now tested the requirement for E2F and pumilio function during regeneration experimentally using RNAi. When the E2F gene upregulated during regeneration is knocked down by RNAi, we do not see any effect on the early stages of oral primordium formation, but the later stages fail to occur. The stages affected correspond to the points in time when the predicted E2F targets are upregulated, which would be consistent with a role for E2F in driving expression of genes required for later stages of regeneration. These new data has been added to Figure 5.

When Pumilio is knocked down, we see an entirely different set of phenotypes – in most cases, the OA forms but is abnormally small. In some cells, no OA forms, and we also see a range of other phenotypes such as misplaced OA, but these are less frequent. It is hard to draw a firm conclusion from these results about the specific functional role of pumilio in regeneration, but at least these experiments indicate that it does play a role. These new data have been added to figure 6.

We believe that this new functional data substantially strengthens the conclusions of this paper, and we thank the reviewer for pushing us in this direction.

The one big aspect of the biology that is missing for this reviewer is any sense of spatial organization. The paper is all about temporal organization. Temporal cascades of gene expression downstream of a perturbation are ubiquitous in Eukaryotes. In that respect, the results of this paper are very familiar. Stentor is extraordinary in its spatial organization, but this aspect of the biology is not addressed. The paper left me desperately wanting to know where the induced genes are transcribed and translated. In particular, to what extent does overall spatial organization arise from self-organization (or directed transport) of proteins post-translation vs local programming of structure by local translation. Addressing that question might require in situ methods that presumably do not yet exist. It is not mentioned in the discussion whether this sort of question been addressed for any ciliate. Some recent methods for addressing local translation do not require in situ hybridization. For example, there are methods for imaging nascent proteins by measuring their proximity to the peptide bond forming site in ribosomes that might be applicable to Stentor.

We share the reviewer's interest in spatial patterning, indeed the reason for studying regeneration in Stentor is in order to learn how cells are able to generate and regenerate spatial patterns. We would like to emphasize, however, that pattern formation is an inherently spatiotemporal process, and thus the temporal aspect is a key element of the story. This is conspicuously the case in Stentor, where the oral apparatus forms through a series of distinct morphological steps, in much the way that a part being machined in a factory will go through a series of distinct shaping steps. Understanding the molecular events at each of these steps therefore seems necessary as a basis on which to further explore the more overtly spatial processes of patterning. A second area where the transcriptional program becomes important is understanding how the cell senses the need to trigger regeneration. A new OA is formed when the old one is removed, but also under a range of physical re-arrangements of the cell that leave the OA still attached, and it is far from obvious how the cells recognizes these disruptions, leaving open a very interesting possibility that the cell is somehow able to sense its own geometry. We are exploring several strategies for identifying the earliest signals that trigger regeneration, but one strategy is to see which genes are triggered first, and then look for known regulators of their expression. In this context, E2F is particularly nice because so much is known about its regulation in other organisms. We therefore believe that the work reported here will, in fact, be invaluable for understanding spatial pattern formation and possibly geometry sensing, even though, as the reviewer correctly points out, our emphasis here is on temporal regulation. We have added the following statement to the introduction to clarify this point:

"In addition to producing lists of candidate genes involved in assembly, the timing with which different genes are expressed will potentially reveal sequential steps in the assembly process itself. Furthermore, by revealing the timing of gene expression during regeneration, transcriptional analysis can provide clues about how the process is initiated. OA regeneration can be triggered by several different procedures, including removal of the OA but also rotation of the OA relative to the rest of the cortical pattern (Tartar 1961) raising the question of how the cell recognizes these geometric perturbations. One approach is to determine which genes are expressed earliest in the pathway, and then move upstream to identify signals required to turn these early-expressed genes on. In terms of the timing and sequential logic of the expression program itself, we note that even if the ultimate goal is to understand how *Stentor* achieves complex spatial patterning, pattern formation is an inherently spatiotemporal process, such that both spatial and temporal control are important."

One way the authors could begin to address spatial regulation without the need for new methods would be to trigger regeneration of the oral apparatus (possibly with and without cycloheximide), then bisect the animal at different time points and collect mRNA from proximal vs distal halves. They could then ask whether induced mRNAs are preferentially transcribed/translated in the half of the organism where the oral apparatus is assembling. It would be important to normalize carefully for total mRNA and ideally to distinguish spatially localized transcription vs mRNA localization. That might be possible by looking at introns. Such studies could be executed in an unbiased way using RNAseq or the authors could look at specific induced genes by qPCR.

We are currently attempting to implement such a strategy. The trick with using our current method is that bisection of Stentor cells is a somewhat random process due to the squirming motion of the cells while one tries to cut them. This means that one essentially has to take a swipe at the cell with the needle and hope that the cut is in approximately the right location. There ends up being significant variation in the precise location of the cut site relative to the equator of the cell. For the type of analysis we report in this manuscript, we don't believe this makes a significant different since we are looking at large scale differences in the expression pattern between the two halves. But for cells in the act of regenerating the OA, which takes place right at the equator in most cells, there would be potentially a large difference between cells bisected slightly anterior to the oral primordium site, versus cells bisected slightly posterior of the site. We have a strategy to handle this by using single-cell RNAseq, in which case the potential variation in cut position from cell to cell can in fact become a feature rather than a bug, by allowing us to infer a spatial ordering by comparing many pairs of bisected halves. To this end, we are developing a workflow for single cell RNAseq for Stentor, so that the pattern of mRNA abundance can be directly compared in each of the two cell-halves resulting from a single bisection experiment. In the long run we believe these experiments will provide a wealth of new information, but we view them as beyond the scope of the current manuscript which, as the reviewer notes, is focused on the temporal program of expression.

The E2F hypothesis is very interesting. The paper would be obviously stronger if this provocative mechanistic hypothesis were tested. The same is true for the pumillio hypothesis. That said, to this reviewer, the issue of spatial regulation is more interesting. Starting to address spatial regulation would add more to the paper than addressing a molecular cause-effect prediction.

As noted above, we have now provided new data about the functional roles of E2F and Pumilio. We agree with the reviewer that spatial regulation is of great interest although, and will be addressing that point in future work, which we believe is beyond the scope of this paper.

In the intro I feel the authors are trying too hard to frame Stentor as a model for cell wounding in general and fail to sufficiently highlight the amazing sub-cellular organization and regeneration capacity of ciliates in general and Stentor in particular. They also seem to blur cellular regeneration and tissue regeneration. This writing style presumably reflects NIH priorities. In this reviewer's opinion, ciliates are not a good model for human anything but will reveal completely new ideas on sub-cellular organization that underlie protist biology.

The reviewer raises a good point – our main motivation for studying Stentor regeneration is that we feel it is an inherently interesting biological process, and that even if its details may differ from how other cells regenerate or undergo morphogenesis, we still hope to learn general principles that might apply across biology. We have now completely re-organized the introduction to emphasize that pattern formation is the primary question and wound healing secondary. We have also tried to better highlight the persistent mystery of Stentor regeneration and how genomic methods may provide an important route to solving it.

We don't necessarily agree that ciliates are not a good model for human anything – certainly past examples such as telomeres and chromatin modification say otherwise. Actually, one of the useful things that we can do now that we have our transcriptional program of regeneration in hand, is we can ask about similarity with other systems. Prompted by another reviewer, we compared gene expression after bisection in Stentor with gene expression in regenerating optic nerves of zebrafish (as a model for single-cell regeneration in an animal) and while the number of genes in common is small (20% of the identifiable Stentor orthologs of differentially expressed zebrafish genes are upregulated during Stentor regeneration) the number is much larger than we can calculate from random chance. In contrast, there is essentially no overlap of genes expressed in Stentor regeneration and genes expressed during regeneration at the tissue scale in flatworms, which isn't really surprising since they are such different processes. We have added this information in the Discussion section.

In general, the transcriptional program data will facilitate other such comparisons. For example, in talking with colleagues who have analyzed transcriptional programs during development of multiciliated epithelia in mammals, we do find some extremely interesting commonalities, including an apparent role of E2F. This is something we are currently collaborating to explore further, and so we do not include this data in the current manuscript. In any case, we thank the reviewer for supporting our motivation to understand the biological mystery of Stentor regeneration regardless of its potential applicability to mammalian biology. We note that a different reviewer criticized us for not devoting enough discussion to potential similarities with animal model systems.

"…..by reducing bicarbonate to CO2" is incorrect chemistry. Should read "by dehydrating carbonic acid to CO2". No redox chemistry is involved.

We thank the reviewer for correcting our mistake. We have changed the text accordingly.

The discussion should touch on the state of genome annotation in Stentor and specify the extent to which this study has improved it. Improving genome annotation is an important contribution and would increase the significance of the paper.

The reviewer raises a good point – in fact the current level of annotation of Stentor genome is quite rudimentary. We have now added this statement to the Discussion section at the end of the section comparing expression in Stentor to other regeneration systems:

"One of the challenges with relating lessons learned in Stentor with other systems is that the annotation of the Stentor genome is still at an early stage. Our results will help to provide annotation information for a subset of genes potentially related to regeneration, but further analysis of gene expression during normal development and during other forms of regeneration will be important to obtain a fuller picture of the Stentor genome."

Once these data are published, we will work with the organizers of ciliate.org to add the transcriptomic information to the annotation of the database. Currently, the annotation of the Stentor genome is indeed quite rudimentary, and providing this information will, we hope, be a step towards improving it.

The comparison of regeneration and cell cycle progression in the discussion was really interesting to this reviewer. This set of points might be improved by adding a figure panel that identified clusters of genes that are common to both processes.

We agree with the reviewer that the cell cycle connection is potentially quite interesting. To investigate this connection further, in addition to our functional studies on E2F, we compared the set of genes upregulated during regeneration in Stentor with a study of genes upregulated at different cell cycle stages in mammalian cells (Kowalczyk et al. 2015). The striking result is that for Stentor genes corresponding to cell cycle regulated mammalian genes, a substantial majority correspond to genes expressed during S phase, and of the remainder, most corresponded to genes expressed during G2M. We have now added this information to the Discussion along with a new supplemental table (Table S10) that lists the genes and their corresponding mouse expression information.

As a side note, it might be interesting to stain Stentor at different stages of regeneration and cell cycle progression with an excellent antibody to Phospho-TP (Cell Signaling Technology #9391S). This antibody brightly stains multiple structures in prophase and metaphase human cells and might reveal interesting aspects of spatial regulation by CDKs in Stentor

We have tried one anti phosphoprotein in Stentor but the result was disappointing in that it brightly stains the entire cell. It is possible that some other fixation condition might allow the antibody to work, but we have often had the experience that antibodies raised against human proteins do not give a clean signal in Stentor, either by immunofluorescence or by Western blotting. We have had much better luck when we raise our own antibodies to Stentor proteins or peptides, partly because we routinely screen pre-immune bleeds from rabbits before inoculation to eliminate those that already make Stentor-reactive antibodies, something that happens a lot with ciliates, supposedly because so many rabbits are infested with ciliated parasites.

We have obtained preliminary phosphoproteomic information from growing Stentor cells, and are now working on obtaining similar data as a function of time during regeneration. We have also tested several kinase and phosphatase inhibitors. We published experiments showing that Aurora kinase inhibitors cause delays in regeneration (Lin 2020), and we have found that other inhibitors seem to affect the contractile behavior. In general, we believe that learning about the spatial pattern of protein phosphorylation in Stentor will be a fruitful approach in the future, but we feel it is beyond the scope of the present paper. Reviewer #2 (Recommendations for the authors):

Stentor coeruleus is a large single-cell ciliate with high regenerative potential, and thus provides the opportunity for researchers to understand how single cells build and regenerate parts. In addition, comparisons with metazoan regenerative models will uncover both deeply conserved principles of regeneration and aspects unique to single cell regeneration. In this study, the authors use RNA-seq to comprehensively profile the transcriptional changes that occur over the course of three regeneration paradigms: (1) Oral apparatus (OA) regeneration that occurs after sucrose shock, (2) Regeneration of the "anterior end" (which includes the OA) after bisection, and (3) Regeneration of the "posterior end" after bisection. By comparing the transcriptional changes that occur during these three distinct regeneration paradigms, the authors were able identify modules that were specific to certain events, such as regeneration of the OA and the reaction to bisection. The authors use these data to identify some putative key regulators. For example, the authors identified E2F as a transcriptional regulator that could be required for OA regeneration, as well as several homologs of the conserved RNA regulator Pumilio with unique expression patterns during regeneration. Notably though, no candidate regeneration regulators were tested using the RNAi methods developed by this laboratory. Finally, RNA-seq was also performed for OA regeneration in the presence of a translation blocker, which allowed the authors to separate transcriptional responses that were initiated by existing proteins (largely the earlier transcriptional responses) from those that were dependent on the production of new proteins. Based on these data, the authors conclude that regeneration occurs as a cascade of transcriptional events, with one wave triggering the next. Although the authors discuss and an alternate regulatory architecture that could conceivably control regeneration in Stentor (the initial injury response individually activating each regeneration module directly), the "cascade" model seems expected in this case. Overall, the study provides a useful and comprehensive starting point to unraveling the molecular mechanism underlying Stentor regeneration. However, three shortcomings of this study are: (1) lack of functional data for identified candidates testing their requirement for regeneration, (2) the finding that regeneration occurs through a "cascade" of transcriptional events seems expected (if it is not expected, the authors would need to cite clarifying literature) and (3) the authors conclude that their data show "deep conservation of developmental mechanisms," which seems strong based on an expression study in which only a small number of possibly conserved genes were discussed in the text.From the perspective of a developmental biologist, it is not surprising that regeneration occurs as a cascade of events (i.e. a GRN) triggered by an immediate early response that is translation-independent. It also doesn't seem surprising that regeneration of different parts would require different gene modules. However, this may be my own limited view. If there are examples from the literature that support the possibility of other regulatory architectures being used in, for example, cellular repair, these should be cited to help support the significance of the findings here.

We agree that a cascade like organization is not in any way a radical or bizarre scheme, given that many developmental systems use this type of organization. But it is clearly not the only possible way to generate a temporal sequence of events – the other alternative, as we indicated, is to have a central timer of some kind that triggers different events as a molecule gradually increases in concentration. In a cell biology context, it has been shown that a single transcription factor can produce a temporally ordered sequence of expression of downstream targets. For example, O'Shea and co-workers found that when the Msn2 transcription factor is induced, target genes whose promotors have a low threshold for activation are transcribed first, and then other target genes with a higher threshold for activation are triggered later. We have added a discussion of this point along with citations to the Discussion section. We note that in Developmental Biology there is a very clear example of a spatial version of this type of system, namely the "French flag" model for pattern formation in which different target genes are induced by different concentrations of a morphogen.

This same question of cascade versus a single timer was a matter of substantial debate in the cell cycle field. Before the basic cell cycle clock was discovered, a viable model was the chain of dominos model, which was essentially a cascade in which each event of cell division would then trigger the next event. This turned out not to be the case at all, and in fact cell division is coordinated not by a cascade but by a centralized production schedule created by the cell cycle clock. We thus do not agree that a cascade is necessarily the obvious choice for regulating the timing of a cellular process, even if it is the more common approach in the context of development in multicellular animals.

It is certainly surprising/confusing that the OA regeneration specific module does not include cilia motility genes. Could an explanation be that it takes longer to regenerate cilia in the bisected animals and the time course didn't go long enough to capture that step?

This is a very interesting suggestion that we had not considered. However, in looking at the existing literature, it is clear that ciliary motility is recovered at the same stage in regeneration whether the OA was removed by sucrose shock or if cells are responding after bisection. We have now added the following statement to the Discussion section:

"Prior studies have found that full ciliary motility, including metachronal waves, is achieved in the oral apparatus at the same stage in development when the oral primordium has reached its full length, regardless of whether regeneration was induced by sucrose shock (Paulin and Bussey 1971) or by surgical cutting (Tartar 1963; James 1967). Thus, the difference that we observe in terms of ciliary gene expression does not reflect a difference in the timing at which ciliary motility is recovered."

Figure 1A,B – It would be helpful to label the Stentor parts and the A/P axis on the figure. Also, is the distribution of micronuclei known? After the bisection, does each half get at least one? Do more need to be made? Can a fragment of stentor regenerate without a micronucleus? These seem like important points in trying to understand how regeneration works. I'm not suggesting experiments should be done to address this point, but if the answers to some of these questions are known, I think it would be helpful for the reader to know.

We have now labelled the diagram in Figure 1A to indicate the anterior and posterior ends of the cell, and to identify the three key components discussed in the text, namely the oral apparatus, the macronucleus, and the holdfast.

The question about micronuclei is a good one, since there are multiple micronuclei, and in bisected cells the multiple micronuclei will get distributed between the two daughters. We do not believe that this will make any difference for the subsequent results since prior studies have found that removal of the micronuclei does not impair cell growth or regeneration, in contrast to the macronucleus, removal of which blocks regeneration entirely. We have now added the following text to the introduction to clarify this point:

"In addition to the macronucleus, each *Stentor* cell contains multiple micronuclei, which are generally not visible because they are closely adjacent to the macronucleus. The micronuclei are required for sexual reproduction but are dispensable for regeneration and for mitotic cell division (Schwartz 1935; Tartar 1961)."

Figure 1C – It would be helpful to define the colors in the figure legend. Also, be consistent with capitalization

We have fixed the capitalization problem with this figure to make it consistent. We also have changed the terminology to reflect the usage in the main text. We have now provided a definition of the circles of the Venn diagram, including their color, in the figure legend.

Figure 2B – What method was used to place genes into the various categories? This information should be included in the methods section.

We have now added a section to Methods entitled "Annotation of subsets of genes" in which we explain how these categories were derived from existing annotation as well as new BLAST searches for the specific centriole-related categories.

Figure 5: Are the putative E2F targets not upregulated in the regenerating tails? If so, this would further support the hypothesis that the list shown in Figure 5B are targets of E2F.

Indeed, as the reviewer predicts, predicted E2F targets are not among the genes upregulated in cells regenerating tail structures. We have now made of a note of this fact in the results:

"In contrast to the identification of multiple predicted E2F target genes among the genes upregulated during regeneration after either sucrose shock or in posterior half cells that are regenerating anterior structures, predicted E2F targets were not observed among genes upregulated during tail regeneration in anterior half cells. "

The methods should include a description of how putative E2F and pumilio binding sites were identified.

We have now added the following information to the Methods:

"In order to identify potential targets of E2F and Pumilio, we searched the non-coding sequence of the Stentor genome using Perl with the following regular expressions:

E2F: TTT[GC][GC]CGC Pumilio: TGTA[CTAG]ATA"

Reviewer #3 (Recommendations for the authors):The manuscript by Sood et al. aims to describe transcriptional changes underlying regeneration in the large, single-celled ciliate Stentor, which is a fascinating and novel model for wound repair, the restoration of polarity, and regeneration. Stentor regeneration likely occurs through mechanisms distinct from those operating in metazoans which rely heavily on differentiation of new cell types. This is a new and exciting model organism for understanding regeneration and the transcriptional programs controlling this process. By comparing different regeneration paradigms (anterior and posterior regeneration, vs sucrose shock which removes only the oral apparatus), the authors aim to obtain a global view of the transcriptional changes that drive regeneration in Stentor. While the biology is potentially interesting, the paper over-relies on transcriptional profiling without any functional testing of the genes, pathways, or processes uncovered.

We thank the reviewer for the positive assessment of the model system and for the goals of our study. We agree that lack of functional evidence was a problem and so we have now provided functional analysis for two key upregulated genes, as discussed below.

1. This manuscript lacks a connection between the transcriptional changes observed during regeneration and any biological outcomes. Without confirmation of these descriptive findings, it is difficult to understand the significance of these newly discovered transcriptional programs in driving regeneration. There are several potential ways that this could be at least attempted: RNAi, previously demonstrated to work in Stentor by the Marshall lab; pharmacological perturbations of key genes or pathways; or (at a bare minimum) clearer grouping of categories according to timing of gene expression or predicted function. The major genes noted in the abstract (E2F and Pumilio) are indeed important in other systems for basic cellular functions, but changes in their expression only hint at their potential function in regeneration.

We have now tested the requirement for E2F and pumilio function during regeneration in Stentor, using RNAi. When the E2F gene upregulated during regeneration is knocked down by RNAi, we do not see any effect on the early stages of oral primordium formation, but the later stages fail to occur. The stages affected correspond to the points in time when the predicted E2F targets are upregulated, which would be consistent with a role for E2F in driving expression of genes required for later stages of regeneration. When Pumilio is knocked down, we see an entirely different set of phenotypes – in most cases, the OA forms but is abnormally small. In some cells, no OA forms, and we also see a range of other phenotypes such as misplaced OA, but these are less frequent. It is hard to draw a firm conclusion from these results about the specific functional role of pumilio in regeneration, but at least these experiments indicate that it does play a role. The new data have been added to figures 5 and 6.

2. Overall, the major conclusion is that regeneration occurs in distinct, transcriptional waves. This phenomenon has been observed in other regenerating systems (e.g. Wenemoser and Reddien, Genes & Dev 2012, which also uses cycloheximide treatment to identify translation-independent transcriptional changes). The analysis and the paper in its current form is overly focused on Stentor, and lacks placement in the context of regeneration. A hugely interesting aspect of studying regeneration in Stentor is that it occurs in single cells – and therefore is unlikely to require the cell division and fate specification that is known to drive regeneration in planarians, axolotls, and mammals. Comparing these differences would be worthwhile and would highlight any similarities that the authors might have observed.

One question that this reviewer's comment raises is whether the genes expressed during Stentor regeneration might overlap with genes expressed during regeneration in multicellular animals. We analyzed the set of genes differentially expressed during regeneration in Planaria from the paper the reviewer suggested, as well as a list of genes differentially expressed during optic nerve regeneration in Zebrafish (using the Regeneration Rosetta database) in order to have examples of multicellular and single cell regeneration in animals. In both cases, the fraction of Stentor orthologs of the differentially expressed genes that were also differentially expressed during Stentor regeneration were not significantly different from that expected by chance. This is not really a surprising result since regeneration of tissues in Planaria involves cell proliferation, migration, and differentiation, none of which happen during Stentor regeneration. Likewise, optic nerve regeneration, while taking place at the single cell level, does not involve formation of cilia or basal bodies, while Stentor regeneration does not involve formation of growth cones or axons, and so it is not surprising that there is little correlation in expression patterns. If, however, we focus only on genes involved in a more general wound response in Stentor (i.e. the "bisection" and "general" clusters), now we do find a statistically significant correlation with the optic nerve regeneration data, however the numbers are small and so we would prefer not to make a strong statement just on this basis. In any case, we have added our analysis in two new supplemental tables (S8 and S9), and we describe the comparisons in the Discussion section.

3. The authors performed 3 different regeneration challenges, presumably to garner the transcriptional changes that are shared between all conditions, and those that are unique to each amputation. Because individual genes in the heat maps are not labeled, it's difficult to glean from the figures which genes fall into the general or specific categories, or are carried through from one figure to the next. Additionally, most of the heat maps are not labeled and therefore do not provide a clear picture of the transcriptional changes that they describe.

It did not prove feasible to list the gene accession numbers in the heatmap figures, because the font size would be so small as to be invisible. Moreover, the accession numbers are not inherently meaningful, whereas gene names are not unambiguously assigned. Instead, we now provide extensive information about each gene corresponding to each of the regeneration clusters as separate tables, which contain not only accession numbers but also information about domains and orthologs in other species. Tables S1-S6 list the genes showing differential expression in the OA cluster, sucrose cluster, regeneration of anterior structures, regeneration of tail structures, bisection-specific, and general regeneration, respectively.

4. In Figure 4C, the text says that 21 genes are unchanged after cycloheximide treatment, but the figure panel has 30 rows. It is unclear whether each row correlates to a single gene. The data presented in this figure is not visually/conceptually connected to the identification of a set of immediate early genes as described in the text. If there is indeed a set of "immediate early" genes why are they not mentioned in the text? Furthermore, the statement that their products are required for subsequent transcriptional programs is not demonstrated by the data in the figure.

Regarding the number of rows in the heatmaps, we note that heatmaps were generated using all transcripts identified, including those that do not have matches in the current set of gene models in StentorDB. This was done in order to have the most possible data to support the temporal structure of the expression clusters. However, the gene lists given in the supplementary tables, as well as our discussion of results, was restricted to those transcripts for which corresponding gene models exist in the genome database. For this reason, the number of rows in each heatmap will in general be slightly larger than the number of genes listed in the corresponding tables. We have now clarified this point in the Methods.

Regarding the immediate early genes, the reviewer makes an interesting point about how to interpret the term "immediate early". We were using the term "immediate early gene" as a conceptual definition, to mean genes that are expressed early in a process and whose expression does not depend on translation of any new proteins. But as the reviewer points out, the term can also be viewed as a list of actual specific genes already defined in different situations such as viral infection or serum response. Obviously, the genes that we have shown to act early in the timecourse and whose expression is not dependent on protein synthesis, do not correspond to the genes that share these same properties in infection or serum response. In order to avoid this confusion, we have removed the term "immediate early" from the results, and we only use it in the Discussion, where we make the analogy more explicit between what we see in Stentor and what is seen in viral infection and serum response. We no longer call the stentor genes "immediate early", now reserving this term for the existing examples from mammalian literature. We hope that this change will resolve any potential confusion about our use of the terminology.

We had originally speculated that the upregulation of these early genes might drive later gene expression, and the reviewer is correct that we did not have any data directly supporting that point. We previously used the phrase "these results suggest" but we have now tempered this by saying "The results suggest a hypothetical cascade model in which one set of genes are directly triggered by a pathway that relies entirely on existing proteins, and then one or more of these gene products trigger the rest of the program, possibly by acting as transcription factors". We hope that the phrase "suggest a hypothetical model" now makes it sufficiently clear that this is a hypothesis and not a proven fact. We do, however, note that our new RNAi results with E2F, one of the early-expressed genes whose transcription does not require translation, does in fact appear to be important for regulating late stages of regeneration.

5. In Figure 5, the logic for selecting E2F is insufficiently conveyed. It is unclear whether E2F one of the more strongly upregulated genes in the cycloheximide-treated animals, or whether the authors are pursuing this as a candidate approach? What are the predicted targets of E2F, how were they identified, and where is the quality control verifying that these targets exist in Stentor? If possible, it would be good to name them in Figure 5B. If they are indeed targets of E2F, then a decrease in their transcription should be observed after cycloheximide treatment.

Our focus on E2F was, as the reviewer suggests, based on taking a candidate approach. We have added the following text to clarify this point:

"Because we found all three members of the Rb-E2F-DP1 module in cluster 2 of the OA specific program, E2F is a promising candidate for a regulatory factor controlling later events in the program. "

Predicted E2F targets were identified by searching the non-coding regions of predicted Stentor genome using a regular expression derived from the known E2F recognition motif. It is thus a strictly bio-informatic criterion. We now provide this information in the Methods.

Regarding the identity of the predicted E2F targets, we now provided the following information:

"The identities of predicted E2F targets among the OA-specific genes are annotated in Supplemental Table S1, which indicates that predicted E2F targets do not fall into any single characteristic functional families. However, consistent with the role of E2F in regulating cyclin transcription in other organisms, the target list in *Stentor* does include cyclin and cyclin associated protein A. "

As the reviewer predicted, all but one of the E2F target failed to be expressed in cells treated with cycloheximide. We have added this information to the results.

6. In Figure 6, there needs to be some analysis to support the significance of genes with pumilio binding sites. The conclusion that pumilio may be regulating signaling pathways because several of these genes are kinases/phosphatases is unconvincing.

We agree that at present our analysis of pumilio targets is based on computational prediction of pumilio recognition motifs. We have now provided information about how these targets were identified in the methods section. But the reviewer's point is well taken, at present we have only a bio-informatic prediction that these really are pumilio targets. Thus, while a number of the predicted targets encode signaling proteins, our claim must be restrained by the nature of our evidence regarding the targets. To this end we have modified our claim to make it clear that this is just a speculation based on the computational predictions:

"which we speculate may potentially suggest a role for Pumilio in regulating signaling pathways"

7. In Figure 2, color coding of clusters should be revised to correlate logically with the message of the figure in 2B/2D (e.g. shades of a single color that get darker or lighter in successive clusters) to show a visual pattern, rather than orange and red representing the earliest/latest clusters.

We see the reviewer's point that having red and orange representing the early and late clusters is confusing. We tried to take the suggestion of using shades of blue to represent the upregulated clusters (2-5) but found it hard to distinguish these clusters from each other. What we have done now is to reserve red for the early cluster of genes that are down-regulated, and then green, light blue, dark blue, and violet representing the clusters 2-5.

8. In Figure 3A/3B, is there a difference between "bisected top" and "top bisection"

We thank the reviewer for noting this inconsistent terminology. We have now modified the label in Figure3B to use "bisected top".

9. In general, the figures do not convey a clear message about specific components of the centrioles, although the results describe them, resulting in confusion about whether this is a major focus of the research or not. For example, Figure 2E would benefit, for the general reader, from a figure diagramming the biological significance of these two graphs, or even titles on top of the graphs to clarify what these different groups of genes represent. Similar for 2H/2I.

We have now modified Figure 2 to provide separate labels for the two graphs that previously were contained in panel E. We have augmented both the text and the figure legend to explain what these groups of genes mean (canonical centriole proteins and conserved cilia-related proteins). Regarding panels 2H/2I (which are now re-labelled to 2I/2J), we have augmented our discussion in the text to make it clear how these panels reflect the groups of genes just discussed in the preceding paragraph.

[Editors’ note: what follows is the authors’ response to the second round of review.]

Reviewer #1 (Recommendations for the authors):The revised manuscript now includes knockdown phenotypes for E2F and Pumilio, which was one of the key deficiencies in the previously submitted version. Despite the addition of these phenotypes, the biological significance of the findings is not connected to the analysis of regeneration in a meaningful way to make a clear conclusion. Overall, many of the conclusions (in particular, whether regeneration relies on a cascade or parallel induction) rely on correlations rather than direct testing of these possibilities.

This is a fair point about the cascade – the cycloheximide experiment is an argument that expression of one or more genes is necessary for expression of another group, but it does not show sufficiency, which is what one would ideally like. We are still working on developing transgenics for this organism, which would provide the tools to test sufficiency. We have been careful in our Discussion to state the results as being "more consistent with a cascade mechanism" in order to clarify this point.

In Figure 4, the impact of cycloheximide on gene expression in all clusters is clear from the heat maps in panel D. The text states that the "largest effect" is in clusters 4 and 5. Can these changes be quantified in some way to make this assessment less subjective? This is important because cluster 1, which is upregulated at the earliest times during regeneration, is also affected, yet the authors make the conclusion that clusters 4 and 5 are most affected by blocking translation. Making this conclusion stronger is essential to attempt to distinguish between the two models in A. Also, what is the impact of cycloheximide treatment on regeneration?

We thank the reviewer for this suggestion, we have now modified the Results section to include more quantitative information about the effect, as follows:

"The average correlation coefficient between cycloheximide treated and untreated genes, for clusters 1-5, were 0.72, 1.54, 2.31, -0.22, and 0.81, respectively. Based on these average correlation coefficients, the largest effects, corresponding to the lowest correlations, were seen in cluster 1, consisting of genes that normally are repressed during regeneration, and in clusters 4 and 5, consisting of genes that normally are upregulated late in regeneration. In the case of cluster 1, there was a general loss of repression when translation was blocked. In the case of clusters 4 and 5, there was an overall reduction in upregulation when translation was blocked. "

Note that we have also clarified that cluster 1 consists of genes that are down-regulated during regeneration.

The last point is also a good one in that we had not explicitly described the effect of cycloheximide on regeneration. We have now added the following text to the Results in which we cite prior studies of cycloheximide treatment in Stentor:

"It has previously been shown that treatment of regenerating Stentor cells with cycloheximide leads to arrest of regeneration if treated within the first 4-5 hours, whereas when cycloheximide is added at later times it causes a delay in the final stages but does not fully prevent regeneration (Burchill 1968; Younger 1972). We note that the time range in which cycloheximide treatment switches from causing arrest to causing a delay is approximately the time at which we see cluster 4 showing its increased gene expression."

Some conclusions in the Results section are difficult to extract from the figures. For example, is there data to back up this statement from the top of p. 15, regarding E2F: "In cells treated with cycloheximide, 12/13 of these targets fail to be expressed." If it is in Figure 5B, it should be clearly marked where the failure is, and at which time, or the gene expression should be assayed in an independent manner.

The information about specific E2F targets and the effect of cycloheximide is given in supplemental Table S1. We have modified the result text to make explicit reference to the information in this table, as follows:

"The identities of predicted E2F targets among the OA-specific genes are annotated in Supplemental Table S1. Based on the data annotated in this table, there are 13 predicted E2F targets, of which only one, SteCoe_2152, showed normal expression when cells were treated with cycloheximide. The other 12/13 of these targets show reduced expression compared to untreated cells. "

In E2F RNAi experiments (Figure 5), the panels should be labeled with time points and RNAi conditions. The methods should include a description of how these phenotypes were scored, because many readers of this paper may not be familiar with how to convincingly identify an OA.

This is an excellent suggestion. We have now added text on the figure to show which panels are control vs RNAi and which panels are 4,6,8, and 24 hours. We have also added to the methods section under RNAi Analysis of E2F and Pumilio, the following explanation of how the OA is scored:

"The presence, position, and size of the OA are assessed by visualizing the membranellar band, which is detected in living cells based on the visualization of a dense array of long, beating cilia. This is in contrast to the body wall cilia which are much shorter and not visible at the magnification used in our imaging. The OA primordium is detected as a flat region on the otherwise curved side of a cell, often with a decreased opacity of the adjacent cytoplasm. High magnification video imaging has shown that this flat patch with a flanking transparent zone is a reliable indicator of the primordium."

The discussion now contains many more broad comparisons of gene expression trends in different contexts and the message is lost. It now reads more like a Results section and would benefit from a strong edit.

We were a little uncertain as to whether this comparison belonged in Results or Discussion, in the end we have kept it in Discussion just to clarify that we are making comparison with published data obtained by others. We do agree with the point about the message being lost, and part of the issue seems to be that we were trying to do two things in this section: compare with other studies of Stentor, and compare with examples from animal regeneration, which are really two very different goals. Consequently, we have now split this section into two sections, entitled "Comparison to other studies of Stentor regeneration" and "Comparison with animal regeneration," respectively. We have made an effort to condense the text to focus on the main points of the comparisons, especially when discussing the comparison with other studies in Stentor and the comparison with planarian genes. We have also tried to provide a "take home message" for each comparison, which hopefully will create more structure in the discussion. These sections are still a bit dense in terms of giving specific numbers, but we don't see a good way to avoid that given the nature of the question being asked. Moving these comparisons to results would be one way, but somehow feels out of place given that the question itself is more in the nature of what we think the Discussion section should be addressing. If the reviewer feels strongly that these parts should go into Results, we would be happy to move them.

Figure 6F-H. The authors should include control images in addition to the RNAi animals. The y axis on H should be labeled. MB should be redefined since it's only mentioned once in the paper.

We have now included control images for the RNAi experiments of Figure 6. We have now added a Y axis label to panel 6H. We have replaced the abbreviation MB in all figures with the alternative term OA, in order to be consistent with the rest of the manuscript.

For all figures where regeneration is measured at different times, the y axis is labeled "regenerating on schedule". I appreciate that some quantification is always arbitrary but there should be some information in the methods about how this subjective measurement was made.

This was an oversight on our part. We have now added the following explanation of how we define "on schedule" to the Methods section on RNAi analysis:

"For the E2F experiments summarized in Figure 5K, regeneration on schedule was defined by the presence of a visible OA primordium on the side of the cell by 4 hours, an OA that contains an elongated membranellar band with visible beating cilia by 6 hours, an OA with motile cilia that has migrated to the anterior end of the cell by 8 hours, and cells that are still alive and have an OA with motile cilia encircling the anterior end. "

We have also clarified in the legend to Figure 5 that regeneration on schedule is defined by a set of milestone events, and we point the reader to the Methods.

At the top of page 11, describing Figure 2G: "Cluster 1 represents down-regulated genes." is confusing, because these genes are initially upregulated but gradually decrease over 7 hours. Please rephrase.

We have added the following statement to the text to indicate that in fact the genes in cluster 1 show their highest expression prior to surgery, and show decreased expression during regeneration.

"Cluster 1 represents down-regulated genes whose expression is highest at t=0, prior to bisection, and then decreases during regeneration."

What makes this somewhat confusing is that the Z statistics are computed for each row, such that the color code indicates deviation from the average expression over all time points. Consequently, if a gene is down-regulated during some time points, those time points will be colored shades of blue, but the zero time point, being higher than the other time points, ends up being colored red. This color scheme seems to be standard in the field, but one does wonder if there is a better way. But if there is, we have not been able to think of it. In any case, we have clarified this point in the figure legend as follows:

"Z-score is calculated per row and color coded with shades of red representing expression higher than the row average and blue lower than the row average. "

Can the defect in OA regeneration observed in E2F RNAi that emerges between 4-6 hours after amputation, and potential E2F targets or genes that are expressed be connected?

The main phenotype is a delay of progression of regeneration, as though the cell is stuck in an earlier stage so that it fails to hit subsequent developmental milestones. We have already speculated in the Discussion that the temporal progression of regeneration may be related to the cell cycle machinery. We think the fact that both a cyclin and a cyclin associated protein are among the E2F targets might potentially be significant in this respect, although at this point it is entirely speculation. We have modified the Discussion section concerning the cell cycle to include the following speculation about E2F targets:

"The fact that knockdown of E2F leads to a delay or failure of late stages of regeneration may in part reflect a role of cell cycle related E2F targets in this system. Indeed, a cyclin, cyclin A associated protein, and Akt kinase are among the predicted E2F targets upregulated during regeneration in Stentor. We speculate that the delay phenotype in E2F RNAi experiments may result from reduced progression through cell cycle transitions associated with successive steps of regeneration."

Reviewer #2 (Recommendations for the authors):One of my previous concerns was the lack of functional data, which has now been added. While these data have the potential to boost the significance of the study, there are issues with the data presentation outlined below:E2F RNAi experiments: The images shown in Figure 5 panels C-J are not sufficiently labeled. It is unclear which panels are RNAi treated and which are controls. It is unclear which time points are shown. It is unclear what the arrow and arrowheads are pointing to. The images are not of high enough quality to make out any detailed features. The text asks the reader to look for the "OA primordium" in the pictures, but there are no discernible structures observed, at least to the untrained eye.

We thank the reviewer for these suggestions in improving the readability of this figure. We have now added text on the figure to show which panels are control vs RNAi and which panels are 4,6,8, and 24 hours. Both the OA, and the OA primordium, are very hard to see in fixed images. We have added an explanation of how we score the presence of a primordium to the methods section as follows:

"The OA primordium is detected as a flat region on the otherwise curved side of a cell, often with a decreased opacity of the adjacent cytoplasm. High magnification video imaging has shown that this flat patch with a flanking transparent zone is a reliable indicator of the primordium."

Pumilio RNAi experiments: The images shown of regenerating cells after pumilio RNAi seem to indicate a disruption or delay in regeneration, but there are issues with data presentation. First, no images of control regenerating cells are shown for comparison. The images are not of high enough quality for a naïve reader to identify the membranellar band (i.e. the structure that is being pointed to in Figure S1F by the purple arrow).

We have now added control images for the RNAi experiment of Figure 6. We have also provided explanation of how the presence of the OA is scored based on visual appearance, as follows:

"The presence, position, and size of the OA are assessed by visualizing the membranellar band, which is detected in living cells based on the visualization of a dense array of long, beating cilia. This is in contrast to the body wall cilia which are much shorter and not visible at the magnification used in our imaging."

Because this identification involves noticing the motion of the membranellar band cilia, it is indeed difficult to see in fixed images. We have tried our best to add arrows to show where these regions are.

In Figure 6H, an orange bar shows up with no explanation in the "spherical cell" category.

We have now changed the name of this category to "round cell" which better reflects the phenotype. An example image is given in Supplementary Figure S1H. We now include a reference to Supplementary Figure S1 within the figure legend of Figure 6 so that the reader can see examples of the various phenotypic categories. We have added the following to the main text section describing the Pumilio RNAi experiments:

"Other, less frequent phenotypes were the presence of two posterior tails (Supplemental Figure S1G), cells having an abnormal rounded appearance rather than the usual elongate Stentor cell shape (Supplemental Figure S1H), and dead cells showing a rupture of the cell and spilling of cytoplasm (Supplemental Figure S1I). "

…the abbreviation "MB" should be explained in the legend (I had to go back up to the introduction to remind myself what that was).

We have replaced the abbreviation MB with the alternative term OA, in order to be consistent with the rest of the manuscript. In order to avoid any confusion, we have added a statement to the Methods to indicate that we score the presence of the OA based on the presence of a visible membranellar band, as follows:

"The presence, position, and size of the OA are assessed by visualizing the membranellar band, which is detected in living cells based on the visualization of a dense array of long, beating cilia. This is in contrast to the body wall cilia which are much shorter and not visible at the magnification used in our imaging."

Beyond the new RNAi data presented, the authors have responded to my other previous concerns satisfactorily.

We are grateful to the reviewer for the constructive suggestions in the last round of review and are glad to hear that our response to those suggestions was satisfactory.